# Unbiased Loss Functions for Multilabel Classification with Missing Labels

**Erik Schultheis**                                                    *erik.schultheis@aalto.fi*
*Aalto University*
*Espoo, Finland*

**Rohit Babbar**                                                          *rb2608@bath.ac.uk*
*University of Bath & Aalto University*
*Bath, UK & Espoo, Finland*

Reviewed on OpenReview: *https://openreview.net/forum?id=hMq1hUhLqp*

## Abstract

This paper considers binary and multilabel classification problems in a setting where labels are missing independently and with a known rate. Missing labels are a ubiquitous phenomenon in extreme multi-label classification (XMC) tasks, such as matching Wikipedia articles to a small subset out of the hundreds of thousands of possible tags, where no human annotator can possibly check the validity of all the negative samples. For this reason, propensity-scored precision—an unbiased estimate for precision-at-k under a known noise model—has become one of the standard metrics in XMC. Few methods take this problem into account already during the training phase, and all of these are limited to loss functions that can be decomposed into a sum of contributions from each individual label. A typical approach to training is to reduce the multilabel problem into a series of binary or multiclass problems, and it has been shown that if the surrogate task should be consistent for optimizing recall, the resulting loss function is not decomposable over labels. Therefore, this paper develops unbiased estimators for generic, potentially non-decomposable loss functions. These estimators suffer from increased variance and may lead to ill-posed optimization problems, which we address by switching to convex upper-bounds. The theoretical considerations are further supplemented by an experimental study showing that the switch to unbiased estimators significantly alters the bias-variance trade-off and may thus require stronger regularization.

## 1 Introduction

Extreme multilabel classification (XMC) is a machine learning setting in which the goal is to predict a small subset of positive (or relevant) labels for each data instance out of a very large (thousands to millions) set of possible labels. Such problems arise for example when annotating large encyclopedia (Dekel and Shamir, 2010; Partalas et al., 2015), in fine-grained image classification (Deng et al., 2010), and next-word prediction (Mikolov et al., 2013). Further applications of XMC are recommendation systems, web-advertising and prediction of related searches in a search engine (Agrawal et al., 2013; Prabhu and Varma, 2014; Jain et al., 2019; Dahiya et al., 2021).

Typical datasets in these scenarios are very large, resulting in possibly billions of (data, label) pairs. This means that it is not possible for human annotators to check each pair, and thus the available training data is likely to contain some errors. Even annotating only a few samples in order to generate a clean test set can be prohibitively expensive. Fortunately, in many cases it is possible to constrain the structure of the labeling errors. Consider, for example, the case of tagging documents. Here, we can assume that each label with which the document has been tagged has been deemed relevant by the annotator, and thus is relatively

surely a correct label. On the other hand, the annotator cannot possibly check hundreds of thousands of negative labels. This leads to the setting of missing labels investigated in this paper, where only positive labels are affected by noise (they can go missing), whereas negative labels remain unchanged (no spurious labels). For a formal definition of the setting we refer the reader to subsection 2.2, and for a more thorough discussion of prior works on missing labels and related settings to section 9.

Many machine learning methods are based on minimizing an expected loss over the data distribution, typically by using the empirical risk as a statistical estimator. Thus, a natural extension to the missing-labels setting is to construct an unbiased estimator of the true risk given noisy data. In the XMC context, such an approach has been introduced by Jain et al. (2016), who constructed *propensity-scored* versions for some common loss functions. They achieve this under the assumption that the probabilities for each label to not go missing (called its propensity) is known, and developed an empirical model to estimate these propensities from data statistics. The model assumes that propensities are identical for every data point (labels go missing independently of features) and contains two dataset-dependent parameters that have only been determined and made available for a few benchmark datasets. Despite these shortcomings, the resulting propensity-scored metrics have found widespread use in the XMC setting (Bhatia et al., 2016).

However, many loss functions that are employed for training, such as binary cross-entropy or the squared-hinge loss, do not fall within the scope of Jain et al. (2016). Consequently, many methods that use propensity-scored precision as an evaluation metric still perform training using a loss function designed for clean-label training (Dahiya et al., 2021; Guo et al., 2019; You et al., 2019; Babbar and Schölkopf, 2019).

Based on the unbiased estimators given in Natarajan et al. (2017) for the setting of class-conditional noise, Qaraei et al. (2021) provide unbiased versions for several common loss functions. Similar to related learning settings (Kiryo et al., 2017; Chou et al., 2020), they observed that the unbiased estimates may be non-convex, non-lower-bounded, and lead to severe overfitting due to high variance. This paper provides some additional analysis in the form of a uniqueness result Theorem 10 that implies that there are no other unbiased estimates with reduced variance, and a generalization bound Theorem 9 which suggests the bias-variance trade-off observed in practice. A mitigation strategy is to interpret the loss functions as convex surrogates of the 0-1 loss, and switch from unbiased estimates of surrogates to convex surrogates of the unbiased estimate (Qaraei et al., 2021; Chou et al., 2020).

Often, XMC problems are formulated as ranking tasks in which the goal is to maximize either recall or precision within the top-k predictions. Instead of optimizing these metrics directly, the task is typically reduced to a series of binary or multiclass problems, with different reductions consistent for either recall or precision Menon et al. (2019).[1] The reductions consistent for precision lead to loss functions that can be decomposed into a sum of contributions from each label which makes them amenable to the methods of Natarajan et al. (2017). In contrast, the reductions consistent for recall contain a normalization term that is the inverse of the total number of true labels. This term is also necessary for calculating the recall metric itself, demonstrating the need for unbiased estimates for true, non-decomposable multilabel loss functions.

The unique, unbiased estimate for the generic multilabel case is provided by Theorem 19. This result can be seen as a special case of Van Rooyen and Williamson (2017, Theorem 5), which states that the unbiased estimate can be calculated by applying the inverse of the adjoint of the label-corruption operator, a $2^l \times 2^l$ matrix for a problem with $l$ possible labels, to the vector of all $2^l$ possible loss values for a given prediction. The solution that comes out of our direct computations requires computation exponential only in the number of observed labels. In large-scale multilabel problems, the number of relevant labels is typically much smaller than that of possible labels (Jain et al., 2016; Busa-Fekete et al., 2022). If it grows logarithmically, then our approach requires only $\mathcal{O}(l)$ evaluations of the loss function.

We derive these results on the basis of modelling the observed labels as the product of the true labels and an (unknown) mask variable. Note that this is different from semi-supervised learning with labelled and unlabelled data, as the mask is only used as a mathematical convenience, but no knowledge of the actual mask values is assumed. The advantage of this formulation is that we can choose a realization of the mask variable such that it is independent of the other random variables, even though the modeled noise

---

[1]While other surrogate loss functions, such as the *multi-label logistic loss* (Mao et al., 2024), these have not yet been scaled to the extremely large label spaces considered here.

is class-conditional. This leads to unbiased estimates that contain products of several propensities in the denominator; for low propensity, or if the number of labels is large, this can lead to very high variance in the estimator.

In order to judge the severity of the variance and overfitting problems in practice, we conducted three experiments. First, in a pure evaluation setting on synthetic, we calculated the unbiased recall@k for a varying fraction of missing labels, which shows that once this fraction becomes too large, the variance of the estimate explodes and it becomes unusable. As a consequence, this metric can only be calculated on datasets with moderate propensities and many data points. The second experiment serves as a demonstration for the change in bias-variance trade-off as a result from switching to unbiased estimates. Here we trained a linear model with varying $L_2$-regularization using the different versions of the loss functions, using real-data with artificially induced missing labels. We find that training with noisy labels generally shifts the trade-off towards higher regularization, and that for the high-variance unbiased estimates of non-decomposable losses, the original version of the loss function works better than the unbiased one. Finally, we repeat that experiment using real data from the Yahoo-Music R3 dataset, which has been sampled in such a way that the proportion of missing labels can be stimated reliably. Unfortunately, the level of noise in this data is so high that the best classifier occurs at extreme regularization, and makes constant predictions independent of any input features. The code for the experiments is provided at `https://github.com/xmc-aalto/missing-labels-tmlr`.

To summarize, the key contributions of this paper are

1. The model for missing labels as a product of true labels and an independent mask variable (Proposition 4), which allows for convenient handling of expectations in proofs and derivations.

2. We first demonstrate its usefulness in the binary case, where we derive new results on uniqueness (Theorem 10) and variance of the unbiased estimators, and provide a generalization bound (Theorem 9) which is a corrected version of Natarajan et al. (2017, Lemma 8).

3. The unbiased estimates for general multilabel functions (Theorem 19). These can be applied to the normalized loss reductions (Menon et al., 2019), which are required for training that is consistent for recall@k. It turns out that even the unbiased estimation of recall@k becomes a very involved procedure (in contrast to precision@k) that requires summing over contributions from all subsets of the observed labels.

4. The investigation of the influence of missing labels and unbiased estimates on the bias-variance trade-off. We confirm the intuition given by the generalization bound that training with the unbiased losses can lead to severe overfitting, requiring a re-tuning of regularization, and in some cases entirely negating the benefit of unbiased estimation. In situations where they are available, convex upper-bounds can be used to mitigate this problem to some degree.

## 2 Preliminaries

### 2.1 Notation

In this paper, random variables will be denoted by capital letters $X, Y, \ldots$, whereas calligraphic letters denote sets and lower case letters their elements, such as $x \in \mathcal{X}$. Vectors will be denoted by bold font, $\boldsymbol{y} \in \mathcal{Y}$, with components $y_1, \ldots, y_k$. The letters $f$, $g$, and $h$ are reserved for functions, $i$, $j$, $k$ denote integers. With $[k]$ we denote the set $\{1, \ldots, k\}$, and $\mathcal{M}(\mathcal{A}, \mathcal{B})$ is the set of all measurable mappings from $\mathcal{A}$ to $\mathcal{B}$.

There are two natural ways to express a multilabel data point for $m$ possible labels: Either as vectors from $\{0, 1\}^m$ or as subsets of $[m]$. We will mostly use the former, and thus define the label space as $\mathcal{Y} = \{0, 1\}^m$. In cases where the subset notation is convenient, we use the symbol $\mathcal{I}(\boldsymbol{v}) \coloneqq \{i \in [m] : v_i = 1\}$ to denote conversion from subset to vector representation, and $\mathbf{1}^{(\mathcal{I}(v))} = v$ for the reverse operation.

Throughout this paper, we assume an abstract probability space $(\Omega, \mathcal{F}, \mathbb{P})$. We further denote with $\mathcal{X}$ the *data space*, $\mathcal{Y}$ the *label space* and $\hat{\mathcal{Y}}$ the *prediction space*. A dataset is defined through the three random

variables $X \in \mathcal{X}$, $\boldsymbol{Y} \in \mathcal{Y}$, and $\boldsymbol{Y}^* \in \mathcal{Y}$, that represent the *data*, *observed label*, and *ground truth label*. We will generally mark quantities pertaining to the unobservable ground-truth with a superscript star and call $(X, \boldsymbol{Y}^*)$ the *clean data*.

For a given loss function $\ell \colon \mathcal{Y} \times \hat{\mathcal{Y}} \longrightarrow \mathbb{R}$, we denote the true and observed risks of a classifier $h \colon \mathcal{X} \longrightarrow \hat{\mathcal{Y}}$ as

$$\mathrm{R}^*_\ell[h] := \mathbb{E}[\ell(\boldsymbol{Y}^*, h(X))], \quad \mathrm{R}_\ell[h] := \mathbb{E}[\ell(\boldsymbol{Y}, h(X))],$$

and mark their empirical counterparts as $\hat{\mathrm{R}}^*_\ell$ and $\hat{\mathrm{R}}_\ell$.

We will derive unbiased versions of functions of the form $f \colon \mathcal{Y} \times \mathcal{Z} \longrightarrow \mathbb{R}$, where $\mathcal{Z}$ can be the prediction space $\hat{\mathcal{Y}}$ (when evaluating a loss function $\ell(\boldsymbol{Y}^*, \hat{\boldsymbol{Y}})$), the data space $\mathcal{X}$ (when evaluating a classifier $\ell(\boldsymbol{Y}^*, h(X))$), or some other form of additional argument to be passed into the function we want to estimate.

## 2.2 Setting

In this paper, we are investigating noisy labels in cases where the noise is such that labels can only go missing. This is described formally by

**Definition 1** (Propensity). *The missing-labels setting we described in the introduction leads to the following conditions on the m random variables*

$$\mathbb{P}[Y_j = 1 \mid Y_j^* = 1, X] =: p_j(X) \qquad \text{labels may go missing,} \tag{1}$$
$$\mathbb{P}[Y_j = 1 \mid Y_j^* = 0, X] = 0 \qquad \text{but no spurious labels.} \tag{2}$$

*The value $p_j(x) \in (0, 1]$ is called the* propensity *of the label $j$ for instance $x$.*

Similar propensity models have been used in extreme classification (Qaraei et al., 2021; Jain et al., 2016; Wydmuch et al., 2021), learning-to-rank (Joachims et al., 2017; Oosterhuis and de Rijke, 2020; Wu et al., 2021), and recommendation systems (Sachdeva et al., 2022; Huang et al., 2020; 2022).

The question remains of how to get the estimates of $\boldsymbol{p}$. In the XMC community, an empirical model proposed by Jain et al. (2016) has become established, which proposes to estimate the propensity of a label $j$ based on the number of its occurrences $n_j$ inside the dataset of $n$ instances. With two dataset-dependent constants $a$ and $b$, setting $c := (\log n - 1)(b + 1)^a$, the proposal is to calculate

$$p_j = (1 + c \exp(-a \log(n_j + b)))^{-1}. \tag{3}$$

It is known that there are some problems associated with this model for the propensity values (Schultheis et al., 2022). The methods presented in this paper are independent of the way the propensity estimates are obtained, so they are applicable also if better propensity values are available, e.g. through a more carefully generated (bias controlled) validation set.

In this setup, we are interested in finding *unbiased versions* of given functions. We want to do this also for functions which take additional arguments, such as the predicted label $Z = \hat{\boldsymbol{Y}}$. Therefore, we need to ensure that $Z$ is *compatible* with the assumed propensity, in the sense of:

**Definition 2** (Compatible with Propensity). *Let $\boldsymbol{p} \colon \mathcal{X} \longrightarrow (0, 1]^m$ be a propensity function, and $(Z, \boldsymbol{Y}^*, \boldsymbol{Y})$ a triple of random variables such that the propensity $p_j(X) = \psi(Z)$ is a function $\psi$ of the second $z$. Iff these variables fulfill*

$$\mathbb{E}[Y_j \mid Y_j^*, Z] = Y_j^* \cdot p_j(X), \tag{4}$$

*we call them* compatible *with $\boldsymbol{p}(\cdot)$, and write $(Z, \boldsymbol{Y}^*, \boldsymbol{Y}) \in \mathcal{P}_{\boldsymbol{p}}(\mathcal{Z})$. When $\mathcal{Z}$ is clear from the context, we drop it in the notation.*

Now we are ready to give precise meaning to the term *unbiased version*:

**Definition 3** (Unbiased Version). *Let $f^* \colon \mathcal{Y} \times \mathcal{Z} \longrightarrow \mathbb{R}$ be a function of labels and an additional argument, and $\boldsymbol{p} \colon \mathcal{X} \longrightarrow (0, 1]^m$ be a propensity function. We call a function $f_{\boldsymbol{p}} \colon \mathcal{Y} \times \mathcal{Z} \longrightarrow \mathbb{R}$ an unbiased version*

*(for propensity $\boldsymbol{p}$) of $f^*$, iff* for all *random variables $(Z, \boldsymbol{Y}^*, \boldsymbol{Y}) \in \mathcal{P}_{\boldsymbol{p}}$ that are compatible with the specified propensity, it holds that*

$$\mathbb{E}[f^*(\boldsymbol{Y}^*, Z)] = \mathbb{E}[f_{\boldsymbol{p}}(\boldsymbol{Y}, Z)]. \tag{5}$$

*Further, we call an operator $\mathfrak{p}_{\boldsymbol{p}} \colon \mathcal{F} \longrightarrow \mathcal{M}(\mathcal{Y} \times \mathcal{Z}, \mathbb{R})$ an* unbiasing operator *for the function set $\mathcal{F} \subset \mathcal{M}(\mathcal{Y} \times \mathcal{Z}, \mathbb{R})$, iff $\mathfrak{p}_{\boldsymbol{p}}(f^*)$ is a unbiased version of $f^*$ for all $f^* \in \mathcal{F}$.*

Note the importance of the *for all random variables* qualifier. Since we a priori do not know the distribution of the data, we need to make sure that the unbiased estimate we calculate will be correct for all possible distributions.

## 2.3 Masking Model

As a first step in deriving the unbiased estimator, we show that there is an equivalent formulation of the problem in which we model the observed labels as the product between some mask and the true labels, $\boldsymbol{Y} = \boldsymbol{M} \odot \boldsymbol{Y}^*$. It turns out that it is always possible to choose $\boldsymbol{M}$ in a convenient way, making it independent of the true labels $\boldsymbol{Y}^*$.

**Proposition 4** (Masking Model). *Let $\boldsymbol{p} \colon \mathcal{X} \longrightarrow (0, 1]^m$ be a propensity function, and $(Z, \boldsymbol{Y}^*, \boldsymbol{Y}) \in \mathcal{P}_{\boldsymbol{p}}$ be a triple of compatible variables, then then there exists a random variable $\boldsymbol{M} \in \{0, 1\}^m$ such that*

$$\boldsymbol{Y} = \boldsymbol{M} \odot \boldsymbol{Y}^* \qquad \text{almost surely}, \tag{6}$$

$$\mathbb{E}\big[\boldsymbol{M} \mid Z, Y_j^*\big] = \boldsymbol{p}(X). \tag{7}$$

*In particular, we can choose $\boldsymbol{M} = (\boldsymbol{1} - \boldsymbol{Y}^*) \odot \boldsymbol{M}' + \boldsymbol{Y}$, with $M_j' \sim \mathrm{Bern}(p_j(X))$ being independent Bernoulli variables.*

*Proof.* Checking (6), we see that $\boldsymbol{M} \odot \boldsymbol{Y}^* = \big((\boldsymbol{1} - \boldsymbol{Y}^*) \odot \boldsymbol{M}' + \boldsymbol{Y}\big) \odot \boldsymbol{Y}^* = \boldsymbol{Y} \odot \boldsymbol{Y}^* = \boldsymbol{Y}$, where the second equality holds because $(\boldsymbol{1} - \boldsymbol{Y}^*) \odot \boldsymbol{Y}^* = \boldsymbol{0}$.

For (7), we get for all $j \in [m]$:

$$
\begin{aligned}
\mathbb{E}\big[M_j \mid Y_j^*, Z\big] &= \mathbb{E}\big[(1 - Y_j^*) \cdot M_j' + Y_j \mid Y_j^*, Z\big] && \text{(total expectation)} \\
&= (1 - Y_j^*)\,\mathbb{E}\big[M_j' \mid Y_j^*, Z\big] + \mathbb{E}\big[Y_j \mid Y_j^*, Z\big] && \text{(measurable factor)} \\
&= (1 - Y_j^*)p_j(X) + Y_j^* \cdot p_j(X) = p_j(X), && \text{(8)}
\end{aligned}
$$

where the last line used the compatibility condition (4). $\qquad\square$

Proposition 4 can be seen as a generalization of a similar statement given in Teisseyre et al. (2020). The independent variables $\boldsymbol{M}$ provide a convenient framework for proving the results that follow, because the independence allows to factorize expectations containing $\boldsymbol{M}$.

## 2.4 Sufficiency of fixed-propensity results

In this section, we show that, for the task of finding an unbiased version of a function $f^*$ for a general propensity $\boldsymbol{p}(x)$ can be reduced to finding an unbiased version for *each* fixed value.

**Proposition 5.** *Let $f^* \colon \mathcal{Y} \times \mathcal{Z} \longrightarrow \mathbb{R}$ be some function such that for any fixed propensity $\boldsymbol{p} \in (0, 1]^m$ an unbiased version is given by $f_{\boldsymbol{p}}$, i.e. $\mathbb{E}[f_{\boldsymbol{p}}(\boldsymbol{Y}, Z)] = \mathbb{E}[f^*(\boldsymbol{Y}^*, Z)]$. Then, for instance-dependent propensity $\boldsymbol{p} \colon \mathcal{X} \longrightarrow (0, 1]^m$, an unbiased version of $f^*$ is given by $f' = f_{\boldsymbol{p}(X)}$ such that*

$$f'(\boldsymbol{y}, z) = f_{\boldsymbol{p}(x)}(\boldsymbol{y}, z). \tag{9}$$

*Proof.* Note that the right-hand-side is well defined, because in Definition 2 we assume the existence of $\psi$ such that $\boldsymbol{p}(x) = \psi(z)$. Using the law of total expectation gives

$$\mathbb{E}[f^*(\boldsymbol{Y}^*, Z)] = \mathbb{E}[\mathbb{E}[f^*(\boldsymbol{Y}^*, Z) \mid Z]] = \mathbb{E}\big[\mathbb{E}\big[f_{\boldsymbol{p}(X)}(\boldsymbol{Y}, Z) \mid Z\big]\big] = \mathbb{E}\big[f_{\boldsymbol{p}(X)}(\boldsymbol{Y}, Z)\big].$$

The middle equality holds because for any fixed $X$, $\boldsymbol{p}(X)$ is a constant. $\qquad\square$

Therefore, we will suppress the dependence of the propensity on the instance in the rest of the paper.

## 3 Unbiased Versions of Binary and Decomposable Functions

In this section, we first consider the problem of binary classification with missing labels. We re-derive known results for unbiased estimates in the notation of this paper. Then, we state a generalization bound that demonstrates a bias-variance trade-off between the unbiased and the original loss, and a uniqueness theorem that shows that this is inevitable for unbiased estimators.

Finally, we show how the binary results can be applied to multilabel cases where the loss decomposes, i.e. can be written as a sum of binary losses. These situations are quite common, as they correspond to applying the *one-vs-all* (OvA) and *pick-all-labels* (PAL) reductions (Menon et al., 2019).

### 3.1 Derivation of Unbiased Binary Losses

We start by investigating the case with only a single label, $m = 1$, leading to a binary classification problem.[2] Therefore, we drop the vector notation and corresponding subscripts for this section.

First, we define a *propensity-scoring* operator that maps a function to a surrogate function that can be used to compensate for missing labels, and prove that this does lead to unbiased versions.

**Definition 6** (PS Operator). *Let $\mathcal{Z}$ be an arbitrary set, $\mathcal{V}$ a vector space, and $f^* \colon \{0,1\} \times \mathcal{Z} \longrightarrow \mathcal{V}$, be some function. Since the first argument can only take the two different values 0 and 1, we can decompose*

$$f^*(y, z) =: y f_+^*(z) + (1 - y) f_-^*(z). \tag{10}$$

*Then, for any $p \in (0, 1]$, we call $f \colon \{0,1\} \times \mathcal{Z} \longrightarrow \mathcal{V}$, $f = \mathfrak{p}_p(f^*)$ defined as*

$$f(y, z) := y \frac{f_+^*(z) + (p - 1) f_-^*(z)}{p} + (1 - y) f_-^*(z) \tag{11}$$

*a* propensity-scored *version of $f^*$, and the mapping $\mathfrak{p}_p \colon f^* \mapsto f$ the* propensity-scoring *(PS) operator for propensity $p$.*

In general, convexity and non-negativeness of $f^*$ need not result in convexity and non-negativeness of $f$, cf. section 3.2.

**Theorem 7** (Unbiased Estimates with Missing Binary Labels). *The PS-operator $\mathfrak{p}_p$ is an unbiasing operator, i.e., for any $(Z, \boldsymbol{Y}^*, \boldsymbol{Y}) \in \mathcal{P}_p$ compatible with $p$, and any $f^* \colon \{0,1\} \times \mathcal{Z} \longrightarrow \mathbb{R}$, the function $f := \mathfrak{p}_p(f^*)$ fulfills*

$$\mathbb{E}[f^*(Y^*, Z)] = \mathbb{E}[f(Y, Z)]. \tag{12}$$

*Proof.* Applying Proposition 4 and extracting measurable factors gives

$$
\begin{aligned}
\mathbb{E}[f(Y, Z) \mid Y^*, Z] &= \mathbb{E}\left[Y \frac{f_+^*(Z) + (p - 1) f_-^*(Z)}{p} + (1 - Y) f_-^*(Z) \,\middle|\, Y^*, Z\right] \\
&= \mathbb{E}[M \mid Y^*, Z] Y^* \frac{f_+^*(Z) + (p - 1) f_-^*(Z)}{p} + (1 - \mathbb{E}[M \mid Y^*, Z] Y^*) f_-^*(Z) \\
&= p Y^* \frac{f_+^*(Z) + (p - 1) f_-^*(Z)}{p} + (1 - p Y^*) f_-^*(Z) = Y^* f_+^*(Z) + (1 - Y^*) f_-^*(Z) \\
&= f^*(Y^*, Z) \,.
\end{aligned}
$$

Taking the expectation of this equation yields the claim. $\qquad\square$

---

[2] Binary as understood in the sense of detecting the presence of absence of some label ("is there a dog in this picture?"), not as a decision between two classes ("does this picture show a dog or a cat?").

This theorem corresponds to Natarajan et al. (2017, Lemma 7), which derives an unbiased loss function for the generic (two-sided) binary label noise setting.

In practice, the argument to the loss function is a prediction given by some classifier $\phi$, and we want to calculate either the risk for this classifier (for evaluation) or a gradient step (for training). If the classifier's weights $\boldsymbol{W} \in \mathbb{R}^k$ do not depend on the data that is being evaluated, e.g. because they were trained on a separate sample, we still get unbiasedness

**Corollary 8.** *Let $\phi \colon \mathcal{X} \times \mathbb{R}^k \longrightarrow \mathbb{R}$ be a binary classifier with $k \in \mathbb{N}$ parameters $\boldsymbol{W}$, and $\ell^* \colon \{0, 1\} \times \mathbb{R} \longrightarrow \mathbb{R}$ be a loss function, $\ell = \mathfrak{p}_p(\ell^*)$. Assume that $(X, Y^*, Y) \in \mathcal{P}_p$, and the network weights are $\boldsymbol{W}$ independent of the data on which we evaluate, $\boldsymbol{W} \perp\!\!\!\perp (X, Y^*, Y)$. Then $(Z, Y^*, Y) \in \mathcal{P}_p$ with $Z = \phi(X, \boldsymbol{W})$, and we have:*

$$\mathbb{E}[\ell^*(Y^*, \phi(X, \boldsymbol{W}))] = \mathbb{E}[\ell(Y, \phi(X, \boldsymbol{W}))] \,,$$
$$\mathbb{E}[\nabla_{\boldsymbol{W}} \ell^*(Y^*, \phi(X, \boldsymbol{W}))] = \mathbb{E}[\nabla_{\boldsymbol{W}} \ell(Y, \phi(X, \boldsymbol{W}))] \,.$$

*Proof.* To show that $(Z, Y^*, Y) \in \mathcal{P}_p$, notice that

$$\mathbb{E}[Y \mid Y^* = 1, \phi(X, \boldsymbol{W})] = \mathbb{E}[\mathbb{E}[Y \mid Y^* = 1, X, \boldsymbol{W}] \mid Y^* = 1, \phi(X, \boldsymbol{W})] \tag{13}$$
$$\text{(independence of } \boldsymbol{W}) \qquad = \mathbb{E}[\mathbb{E}[Y \mid Y^* = 1, X] \mid Y^* = 1, \phi(X, \boldsymbol{W})] \tag{14}$$
$$((X, Y^*, Y) \in \mathcal{P}_p) \qquad = \mathbb{E}[p \mid Y^* = 1, \phi(X, \boldsymbol{W})] = p \,. \tag{15}$$

Consequently, the first statement follows from a direct application of Theorem 7, and the second because $\mathfrak{p}_p$ is a linear operator, and thus commutes with the gradient, $\nabla \mathfrak{p}_p(\ell^*) = \nabla \mathfrak{p}_p(\ell^*)$. $\qquad \square$

### 3.2 Examples

**Squared Error** For the squared error $\ell^*_{\text{SE}}(y, \hat{y}) = (y - \hat{y})^2$ the unbiased estimate is

$$\ell_{\text{SE}}(y, \hat{y}) = \frac{y}{p}(1 - 2\hat{y}) + \hat{y}^2, \tag{16}$$

where we used $y^2 = y$ because $y \in \{0, 1\}$. In case the label is present ($y = 1$), this is minimized for $\hat{y} = \frac{y}{p}$. The minimum is outside the interval $[0, 1]$, as shown in Figure 1.

**Binary Cross-Entropy** The BCE loss is given by

$$\ell^*_{\text{BCE}}(y, \hat{y}) = -y \log(\hat{y}) - (1 - y) \log(1 - \hat{y}). \tag{17}$$

Since this is linear in $y$, we can directly write down the unbiased estimate as

$$\ell_{\text{BCE}}(y, \hat{y}) = -\frac{y}{p} \log(\hat{y}) - \left(1 - \frac{y}{p}\right) \log(1 - \hat{y}). \tag{18}$$

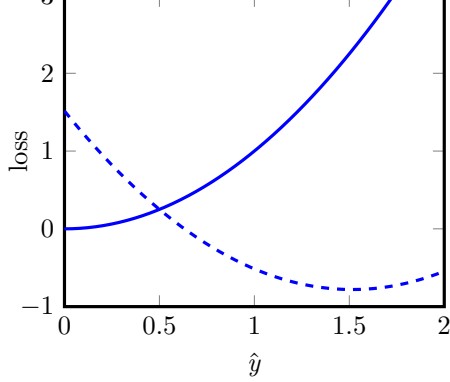

Figure 1: PS squared error loss for $p = 0.66$. The dashed line denote the loss for $y = 1$, the solid one for $y = 0$.

Note that, since the BCE loss is not upper-bounded, the unbiased BCE loss is not lower-bounded, which may be bad from an optimization perspective.

### 3.3 Properties of the Unbiased Estimator

Even though the unbiased estimate guarantees to give correct results in the case of unlimited data, it is not directly clear how helpful it is in the finite data regime. In the related problems of PU-learning, learning from complementary labels, and counterfactual estimation, unbiased estimators are known but they have been found to be problematic in practice (Kiryo et al., 2017; Chou et al., 2020; Swaminathan and Joachims, 2015).

A first problem is that even if the original loss has been chosen as a convex function, the unbiased estimator may not be convex. A sufficient condition for convexity is given in Natarajan et al. (2017, Lemma 10), but as the previous example indicates, many loss functions of practical interest are not of this type. In fact, if the original loss is not upper-bounded (e.g. hinge-loss, logistics loss, squared loss), then the unbiased estimate may not even be lower-bounded, thus rendering the optimization-problem ill-defined. A way of addressing this problem is to forgo the unbiasedness and switch to convex surrogate losses as discussed in section 6.

**Variance**   A second potential problem with unbiased estimators is their variance. In the regime of $p \to 0$, we can show that in the binary case the variance grows with $p^{-1}$ compared to the evaluation on clean data. For the noiseless case, the variance for a fixed prediction $\hat{y}$ is given by

$$
\begin{aligned}
\mathbb{V}[f^*(Y^*, \hat{y})] &= \mathbb{V}\left[Y^* f_+^*(\hat{y}) + (1 - Y^*) f_-^*(\hat{y})\right] \\
&= \mathbb{V}[Y^*]\left(f_+^*(\hat{y}) - f_-^*(\hat{y})\right)^2
\end{aligned}
\tag{19}
$$

whereas the noisy estimator has a variance of

$$
\begin{aligned}
\mathbb{V}[f(Y, \hat{y})] &= \mathbb{V}\left[Y \frac{f_+^*(\hat{y}) + (p - 1) f_-^*(\hat{y})}{p} + (1 - Y) f_-^*(\hat{y})\right] \\
&= \mathbb{V}[Y]\left(\frac{f_+^*(\hat{y}) + (p - 1) f_-^*(\hat{y})}{p}\right)^2 + \mathbb{V}[Y] f_-^*(\hat{y})^2 \\
&= \mathbb{V}[Y]\frac{\left(f_+^*(\hat{y}) + (p - 1) f_-^*(\hat{y})\right)^2 + p^2 f_-^*(\hat{y})^2}{p^2}.
\end{aligned}
\tag{20}
$$

For propensities much smaller than 1, this can be approximated by (setting $q := \mathbb{E}[Y]$)

$$
\approx \mathbb{V}[Y]\frac{\left(f_+^*(\hat{y}) - f_-^*(\hat{y})\right)^2}{p^2} = q(1 - q)\frac{\left(f_+^*(\hat{y}) - f_-^*(\hat{y})\right)^2}{p^2}.
\tag{21}
$$

Setting $q^* := \mathbb{E}[Y^*] = q/p$, and using $1 - q = 1 - pq^* \approx 1$ we get

$$
\mathbb{V}[f(Y, \hat{y})] \approx \frac{q(1 - q)}{p^2}\frac{\mathbb{V}[f^*(Y^*, \hat{y})]}{q^*(1 - q^*)} \approx \frac{1}{p(1 - q^*)}\mathbb{V}[f^*(Y^*, \hat{y})],
\tag{22}
$$

which means that the variance increases linearly with inverse propensity.

**Generalization**   The preceding argument indicates that there might be a bias-variance trade-off between using the unbiased loss that may overfit more strongly on the observed noise, and using the original loss function which gives wrong results even if $n \to \infty$. A first step toward understanding this is to determine upper bounds on the generalization errors (proof in appendix B.2):

**Theorem 9** (Generalization bounds). *Let $\mathcal{H}$ be a function class with Rademacher complexity $\mathfrak{R}_n(\mathcal{H})$. Let $f^*\colon \mathcal{Y} \times \hat{\mathcal{Y}} \longrightarrow [a, b]$ for $a < b \in \mathbb{R}$ be a function that is $\rho-$Lipschitz continuous in its second argument. Let $f := \mathfrak{p}_p(f^*)$ and denote*

$$
r^\star := \inf_{h \in \mathcal{H}} \mathrm{R}_{f^*}^*[h], \quad \hat{h} := \underset{h \in \mathcal{H}}{\operatorname{argmin}} \, \hat{\mathrm{R}}_f[h], \quad \tilde{h} := \underset{h \in \mathcal{H}}{\operatorname{argmin}} \, \hat{\mathrm{R}}_{f^*}[h].
\tag{23}
$$

*as well as*

$$
\begin{aligned}
c &:= \rho \, \mathfrak{R}_n(\mathcal{H}) + (b - a)\sqrt{\frac{\log(2/\delta)}{2n}} \\
m &:= \sup_{z \in \mathcal{Z}}(|f^*(1, z) - f^*(0, z)|).
\end{aligned}
\tag{24}
$$

*For a given sample of $n$ points, it holds with probability at least $1 - \delta$*

$$
\hat{r} := \mathrm{R}_{f^*}^*\left[\hat{h}\right] \qquad\qquad \leq r^\star \qquad\qquad\qquad\qquad\qquad +2\frac{2 - p}{p}c
\tag{25}
$$

$$
\tilde{r} := \mathrm{R}_{f^*}^*\left[\tilde{h}\right] \qquad\qquad \leq r^\star + \qquad\qquad q\frac{1 - p}{p}m \qquad\quad +2c,
\tag{26}
$$

*where $q := \mathbb{E}[Y] \leq 1$.*

Given a hypotheses class $\mathcal{H}$, the (expected) *Rademacher complexity* for $n$ sample points is defined by

$$\mathfrak{R}_n(\mathcal{H}) = \mathbb{E}\left[\sup_{h \in \mathcal{H}} \sum_{i=1}^{n} \sigma_i h(Z_i)\right], \tag{27}$$

where $\sigma_i \in \{-1, 1\}$ are independent Rademacher variables (cf. Shalev-Shwartz and Ben-David (2014, Ch. 26)).

The bound on the original loss function has, in addition to the Bayes and approximation errors error $r^\star$, a second bias term $q\frac{1-p}{p}m$ that does not decrease as $n$ increases. On the other hand, the unbiased estimation introduces a factor of $\frac{2-p}{p}$ in front of the sample-size-dependent term. This indicates that, if only few training samples are available, the ERM of the unbiased loss might result in a worse classifier. This so-called *propensity-overfitting* has also been observed in counterfactual estimation (Swaminathan and Joachims, 2015).

**Uniqueness**   The results above raise the question whether there might be other unbiased estimators with reduced variance and better generalization performance. For example, the conditional expectation $\mathbb{E}[f^*(Y^*, Z) \mid Y]$ also gives an unbiased estimate with lower variance, but cannot be calculated without knowledge of the conditional probabilities $\mathbb{P}\{Y \mid Z\}$. The following theorem states that $\mathfrak{p}_p$ is essentially unique if the marginal probability of the label is known beforehand, and completely unique otherwise.

**Theorem 10** (Uniqueness). *Let $|\mathcal{Z}| \geq 2$, $p \in (0, 1]$, $q \in (0, p]$, and $\mathcal{F} = \mathcal{M}(\{0, 1\} \times \mathcal{Z}, \mathbb{R})$ be a set of functions. Let $\mathfrak{p} \colon \mathcal{F} \longrightarrow \mathcal{F}$ be an operator that maps a function to an unbiased estimate in the missing labels setting, such that for all $(Z, Y, Y^*) \in \mathcal{P}_p$ that fulfill the masking model with propensity $p$ and marginal $q = \mathbb{E}[Y]$, it holds*

$$\forall f^* \in \mathcal{F} : \mathbb{E}[f^*(Y^*, Z)] = \mathbb{E}[\mathfrak{p}(f^*)(Y, Z)]. \tag{28}$$

*Then, $\mathfrak{p}$ is related to the propensity scoring operator $\mathfrak{p}_p$ by*

$$\mathfrak{p}(f^*)(y, z) = \mathfrak{p}_p(f^*)(y, z) + (y - q)\gamma \tag{29}$$

*for some $\gamma \in \mathbb{R}$.*

*Proof.* The sufficiency follows from Theorem 7, the necessity can be shown by choosing distributions of $X, Y^*$ in which $X$ is concentrated on two points. Details in B.1. $\qquad\square$

Since the $(y - q)\gamma$ term cannot depend on the predictions, it is irrelevant from an optimization perspective, and thus the the propensity-scoring operator $\mathfrak{p}_p$ is essentially the only unbiasing operator.

## 3.4   Multilabel Losses that Decompose to Binary Losses

By linearity of the expectation, we can trivially extend the results above to multilabel functions of the following form:

**Definition 11** (Decomposable Function). *A multilabel function $f^* \colon \mathcal{Y} \times \mathcal{Z} \longrightarrow \mathbb{R}$ is called* decomposable, *iff it can be written as*

$$f^*(\boldsymbol{y}, z) = \sum_{i=1}^{m} f_i^*(y_i, z), \tag{30}$$

*with $f_i^* \colon \{0, 1\} \times \mathcal{Z} \longrightarrow \mathbb{R}$. We denote the set of all decomposable functions with $\mathcal{F}_\mathrm{d} \subset \mathcal{M}(\mathcal{Y} \times \mathcal{Z}, \mathbb{R})$. All other multilabel functions are called* non-decomposable.

For decomposable loss functions, we can formulate the following

**Corollary 12** (Unbiased Version for Decomposable Functions). *Let $f^* \in \mathcal{F}_\mathrm{d}$ be a decomposable multilabel function with constituent functions $f_i^*$ and corresponding propensity scored functions $f_i := \mathfrak{p}_{p_i}(f_i^*)$. Then its unbiased version is given by*

$$f(\boldsymbol{y}, z) := \sum_{i=1}^{m} f_i(y_i, z). \tag{31}$$

*In particular, the debiasing operator* $\mathfrak{p}_{\boldsymbol{p}} \colon \mathcal{F}_{\mathrm{d}} \longrightarrow \mathcal{F}_{\mathrm{d}}$ *preserves linear decomposability.*

This corollary allows us to write down unbiased versions for all loss functions that result from the *One-vs-All* and *Pick-all-Labels* reductions, as explained below.

### 3.5 Multilabel Reductions

A common goal in large-scale multilabel tasks is to optimize the *precision at the top*, i.e., the classifiers proposed $k$ labels, and the fraction of truly relevant ones should be maximized. This is the *precision-at-k* metric, defined through

$$\mathrm{P@}k(\boldsymbol{y}, \hat{\boldsymbol{y}}) \coloneqq k^{-1} \sum_{j \in \mathrm{top}_k(\hat{\boldsymbol{y}})} y_j \hat{y}_j \,, \tag{32}$$

where $\mathrm{top}_k(\hat{\boldsymbol{y}})$ returns the set of the $k$ indices with highest scores in $\hat{\boldsymbol{y}}$. This objective is decomposable, and Corollary 12 yields the *propensity-scored precision*(Jain et al., 2016) as its unbiased version,

$$\mathrm{P@}k(\boldsymbol{y}, \hat{\boldsymbol{y}}) \coloneqq k^{-1} \sum_{j \in \mathrm{top}_k(\hat{\boldsymbol{y}})} p_j^{-1} y_j \hat{y}_j \,. \tag{33}$$

As precision-at-k is a non-continuous function, optimizing it directly is a difficult task. Instead, the typical approach is to *reduce* the multilabel problem into a set of easier problems with known solution strategies. The two most common reductions are *One-vs-All* (OVA), which results in binary problems, and *Pick-all-Labels* (PAL), which results in multiclass problems. Let $\ell_{\mathrm{BC}}^* \colon \{0, 1\} \times \mathbb{R} \longrightarrow \mathbb{R}$ be a binary loss, then the corrsponding OVA loss is given by

$$\ell_{\mathrm{OVA}}^*(\boldsymbol{y}^*, \hat{\boldsymbol{y}}) = \sum_{j=1}^{m} \ell_{\mathrm{BC}}^*(y_j^*, \hat{y}_j) \,. \tag{34}$$

In contrast, PAL considers all the positive labels for each instance and tries to minimize their corresponding multiclass loss $\ell_{\mathrm{MC}}^* \colon [m] \times \mathbb{R}^m \longrightarrow \mathbb{R}$, leading to

$$\ell_{\mathrm{PAL}}^*(\boldsymbol{y}^*, \hat{\boldsymbol{y}}) = \sum_{j=1}^{m} y_j^* \ell_{\mathrm{MC}}^*(j, \hat{\boldsymbol{y}}) \,. \tag{35}$$

If $\ell_{\mathrm{BC}}$ and $\ell_{\mathrm{MC}}$ are strictly proper losses, these approaches are consistent for precision at $k$ (Menon et al., 2019).

As a slight generalization, to admit e.g. weighted versions to address the imbalance in the label distribution, we allow the functions $\ell_{\mathrm{BC}}$ and $\ell_{\mathrm{MC}}$ to be different for different labels. This leads to the following two Corollaries:

**Corollary 13** (One-vs-All)**.** *For the one-vs-all reduction with binary losses* $\{\ell_{\mathrm{BC}}^{*\,j}\}_{j=1}^m$, *such that* $\ell^*(\boldsymbol{y}^*, \hat{\boldsymbol{y}}) = \sum_{j=1}^{m} \ell_{\mathrm{BC}}^{*\,j}(y_j^*, \hat{y}_j)$, *an unbiased version is given by*

$$\mathfrak{p}_{\boldsymbol{p}}(\ell^*) = \sum_{j=1}^{m} \mathfrak{p}_{p_j}(\ell_{\mathrm{BC}}^{*\,j}) = \ell \,; \quad \ell(\boldsymbol{y}, \hat{\boldsymbol{y}}) \coloneqq \sum_{j=1}^{m} \mathfrak{p}_{p_j}\left(\ell_{\mathrm{BC}}^{*\,j}\right)(y_j, \hat{y}_j). \tag{36}$$

**Corollary 14** (Pick-all-Labels)**.** *For a PAL loss function*

$$\ell^*(\boldsymbol{y}^*, \hat{\boldsymbol{y}}) \coloneqq \sum_{j=1}^{m} y_j^* \ell_{\mathrm{MC}}^j(\hat{\boldsymbol{y}}), \tag{37}$$

*the unbiased version is given by*

$$\ell(\mathbf{y}, \hat{\boldsymbol{y}}) \coloneqq \sum_{i=1}^{m} \frac{y_i}{p_i} \ell_{\mathrm{MC}}^j(\hat{\boldsymbol{y}}). \tag{38}$$

This means that in the PAL setting, the unbiased version is just a weighted sum of the original constituent functions. In particular, if these are convex (bounded), then the unbiased version is also convex (bounded).

## 4 Unbiased Versions of Pairwise-Decomposable Functions

Among the non-decomposable functions, there is an important subgroup that is still easily tractable within the given framework: Functions in which the dependence on the labels is limited to pairwise interactions. These functions also serve as a simpler setting in which we can introduce the techniques used later in Section 5 to derive the general formula for unbiased versions of non-decomposable functions.

Their are four options for the state each pair of labels can be in, so the most general from of the pair-based functions can be written as

$$f^*(\boldsymbol{y}, z) = \sum_{i=1}^{m} \sum_{j=i}^{m} g_{ij}^*(y_i, y_j, z) \tag{39}$$

for functions $g_{ij}^* \colon \mathcal{Y} \times \mathcal{Y} \times \mathcal{Z} \longrightarrow \mathbb{R}$.

**Refining our assumptions**   Because this contains terms in which multiple labels interact, Definition 2 is not sufficient to ensure the existence of an unbiased version, as it does not prohibit one label $Y_i^*$ going missing depending on the presence of other labels $Y_j^*$. We thus need to amend this definition to

**Definition 15** (Jointly Compatible). *Let $(Z, \boldsymbol{Y}^*, \boldsymbol{Y}) \in \mathcal{P}_{\boldsymbol{p}}(\mathcal{Z})$ for a propensity function $\boldsymbol{p} \colon \mathcal{X} \longrightarrow (0, 1]^m$. Iff the observed labels form a set of independent variables, conditioned on the true labels and the side information, i.e.,*

$$\mathbb{P}[\boldsymbol{Y} = \boldsymbol{v} \mid \boldsymbol{Y}^*, Z] = \prod_{j=1}^{m} \mathbb{P}[Y_j = v_j \mid Y_j^*, Z], \tag{40}$$

*then we call them* jointly compatible *with $\boldsymbol{p}(\cdot)$, and write $(Z, \boldsymbol{Y}^*, \boldsymbol{Y}) \in \mathcal{P}_{\boldsymbol{p}}^{\perp\!\!\!\perp}(\mathcal{Z}) \subset \mathcal{P}_{\boldsymbol{p}}(\mathcal{Z})$.*

It is important to stress that we only require the labels to *go missing independently*, not that the ground-truth labels themselves are conditionally independent given the instance.

**Lemma 16.** *Let $(Z, \boldsymbol{Y}^*, \boldsymbol{Y}) \in \mathcal{P}_{\boldsymbol{p}}^{\perp\!\!\!\perp}(\mathcal{Z})$, and $\boldsymbol{M}$ be the random mask variable constructed in Proposition 4. Then the individual masks are independent, conditioned on the true labels and the auxiliary $Z$, and in particular, with $\boldsymbol{M}_{\neg j}$ denoting all but the j'th component,*

$$\mathbb{E}[M_j \mid \boldsymbol{Y}^*, \boldsymbol{M}_{\neg j}, Z] = p_j(X). \tag{41}$$

*Proof.* We first plug in the definition of $\boldsymbol{M}$ from Proposition 4 and extract measurable factors, then we apply Definition 15:

$$\mathbb{E}[M_j \mid \boldsymbol{Y}^*, \boldsymbol{M}_{\neg j}, Z] = \mathbb{E}\big[(1 - Y_j^*)M_j' + Y_j \mid \boldsymbol{Y}^*, \boldsymbol{Y}_{\neg j}, Z\big] \tag{42}$$

$$= (1 - Y_j^*)p_j(X) + \mathbb{E}[Y_j \mid \boldsymbol{Y}^*, \boldsymbol{Y}_{\neg j}, Z] = (1 - Y_j^*)p_j(X) + \mathbb{E}[Y_j \mid \boldsymbol{Y}^*, Z] \tag{43}$$

$$= (1 - Y_j^*)p_j(X) + Y_j^* p_j(X) = p_j(X). \tag{44}$$

For the last line we used Definition 2.   $\square$

**Deriving an unbiased version**   Next, we show that the expression (39) can be written as a *linear* function in each of the observed labels:

**Lemma 17.** *Let $g^* \colon \{0, 1\} \times \{0, 1\} \times \mathcal{Z} \longrightarrow \mathbb{R}$, then we can write this as a linear function*

$$g^*(y_i, y_j, z) = \sum_{a=0}^{1} \sum_{b=0}^{1} (y_i(2a - 1) + 1 - a)(y_j(2b - 1) + 1 - b) g^*(a, b, z). \tag{45}$$

*Proof.* We achieve this by summing over all four options for $(y_i, y_j) \in \{(0, 0), (1, 0), (0, 1), (1, 1)\}$, and selecting the appropriate one using an indicator function. The expression thus becomes

$$g^*(y_i, y_j, z) = \sum_{a,b=0}^{1} \mathbb{1}\{y_i = a, y_j = b\} g^*(a, b, z). \tag{46}$$

The indicator $\mathbb{1}\{y_i = a, y_j = b\}$, in turn, can be written as the product $\mathbb{1}\{y_i = a\} \cdot \mathbb{1}\{y_j = b\}$. Plugging in $\mathbb{1}\{y_i = a\} = (2a-1)y_i + 1 - a$ yields the claim. $\qquad\square$

With this, we can derive the unbiased version of all pairwise loss functions

**Theorem 18.** *Let* $(Z, \boldsymbol{Y}^*, \boldsymbol{Y}) \in \mathcal{P}_{\boldsymbol{p}}^{\perp\!\!\!\perp}(\mathcal{Z})$, *and* $g^* \colon \mathcal{Y} \times \mathcal{Y} \times \mathcal{Z} \longrightarrow \mathbb{R}$ *as in* (39). *Then an unbiased version of* $g^*$ *is given by*

$$g(\boldsymbol{y}, z) = \sum_{i=1}^{m} \sum_{j=i}^{m} g_{ij}(y_i, y_j) \tag{47}$$

$$\text{with} \quad g_{ij}(y_i, y_j) = \sum_{a,b=0}^{1} \left( (2a-1)\frac{y_i}{p_i} + 1 - a \right) \cdot \left( (2b-1)\frac{y_j}{p_j} + 1 - b \right) g_{ij}^*(a, b, z) \tag{48}$$

$$\text{and} \quad g_{ii}(y_i, y_i) = \left( 1 - \frac{y_i}{p_i} \right) g_{ii}^*(0, 0, z) + \left( \frac{y_i}{p_i} \right) g_{ii}^*(1, 1, z). \tag{49}$$

*Proof.* For the diagonal elements, $g_{ii}^*$ is a function of a single label, and thus Theorem 7 can be applied. For the case $i \neq j$, using the law of total probability and pull-out of measurable factors, as well as Lemma 16:

$$\mathbb{E}[g_{ij}(Y_i, Y_j, Z) \mid Z, \boldsymbol{Y}^*] =$$

$$\sum_{a,b=0}^{1} \mathbb{E}\left[ \left( (2a-1)\frac{M_i Y_i^*}{p_i} + 1 - a \right) \cdot \left( (2b-1)\frac{M_j Y_j^*}{p_j} + 1 - b \right) \,\middle|\, Z, \boldsymbol{Y}^* \right] g_{ij}^*(a, b, Z) =$$

$$\sum_{a,b=0}^{1} \left( (2a-1)\frac{\mathbb{E}[M_i \mid Z, \boldsymbol{Y}^*]Y_i^*}{p_i} + 1 - a \right) \cdot \left( (2b-1)\frac{\mathbb{E}[M_j \mid Z, \boldsymbol{Y}^*]Y_j^*}{p_j} + 1 - b \right) g_{ij}^*(a, b, Z)$$

$\qquad\square$

**Example** An example for a pairwise loss is the Kendall-Tau loss (Shalev-Shwartz and Ben-David, 2014, p. 202), which is used for ranking applications and counts how many pairs of labels are ranked differently in the prediction than in the ground truth. Multilabel classification can be interpreted as a form of bipartite ranking, where no loss is incurred if both labels have the same ground-truth independent of their predictions (Bucak et al., 2009). In that case, we have $g_{ii}^* = 0$ and

$$g_{ij}^*(0, 0, \hat{\boldsymbol{y}}) = 0 \qquad\qquad g_{ij}^*(1, 0, \hat{\boldsymbol{y}}) = \mathbb{1}\{\hat{y}_i < \hat{y}_j\} \tag{50}$$

$$g_{ij}^*(0, 1, \hat{\boldsymbol{y}}) = \mathbb{1}\{\hat{y}_i > \hat{y}_j\} \qquad\qquad g_{ij}^*(1, 1, \hat{\boldsymbol{y}}) = 0. \tag{51}$$

Using the zeros for equal ground-truth ranking, the unbiased estimate can be simplified to

$$\text{KT}(\boldsymbol{y}, \hat{\boldsymbol{y}}) = \sum_{i=1}^{l} \sum_{j=i+1}^{l} p_i^{-1} p_j^{-1} \left( p_i - y_i \right) y_j \mathbb{1}\{\hat{y}_j < \hat{y}_i\} + y_i \left( p_j - y_j \right) \mathbb{1}\{\hat{y}_i < \hat{y}_j\}. \tag{52}$$

For training, instead of the non-differentiable indicator $\mathbb{1}\{\hat{y}_j < \hat{y}_i\}$, one typically uses a margin function $\max(0, \hat{y}_i - \hat{y}_j + 1)$.

## 5 Unbiased Versions of Non-decomposable Multilabel Functions

In this section, we consider the fully generic multilabel case. We first derive the general formula for determining the unbiased version and show some of its properties. Then we define a generic form of *normalized reductions* and apply the formula. Finally, we provide some concrete examples of functions with their corresponding unbiased versions.

| Dataset | Avg. Labels | Max. Labels | Loss evals | > 12 | Capped |
|---|---|---|---|---|---|
| Eurlex | 5.3 | 24 | 1173 | 0.1% | 65 |
| AmazonCat-13k | 5.0 | 57 | $1.2 \times 10^{12}$ | 4.9% | 148 |
| Amazon-670k | 5.5 | 7 | 69 | 0.0% | 69 |
| Wiki-500k | 4.7 | 274 | $1.7 \times 10^{76}$ | 5.2% | 156 |
| Amazon-3M | 36.1 | 100 | $1.0 \times 10^{6}$ | 60.7% | 344 |

Table 1: Dataset statistics, showing average and maximum labels per instances, as well as the number of required original $f^*$ loss evaluations for (53). $> 12$ indicates the fraction of points with more than 12 positives, Capped the required loss evaluations if such points were ignored.

## 5.1 Generic Multilabel Case

For the general multilabel case, we can state:

**Theorem 19** (Multilabel Loss). *Let $(Z, \boldsymbol{Y}^*, \boldsymbol{Y}) \in \mathcal{P}_{\boldsymbol{p}}^{\perp\!\!\!\perp}(\mathcal{Z})$, and $f^*: \mathcal{Y}^m \times \mathcal{Z} \longrightarrow \mathbb{R}$. An unbiased version of $f^*$ can be calculated using the propensity-scored expression $\mathfrak{p}_{\boldsymbol{p}}(f^*) = f$ given by*

$$f: \ (\boldsymbol{y}, z) \mapsto \sum_{\boldsymbol{y}' \preceq \boldsymbol{y}} \left( \prod_{j:y_j=1} \frac{y_j'(2 - p_j) + p_j - 1}{p_j} \right) f^*(\boldsymbol{y}', z) \tag{53}$$

*where $\boldsymbol{y}' \preceq \boldsymbol{y}$ means $\{0,1\} \ni y_j' \leq y_j$.*

*Proof.* Decompose the function $f^*$ into contributions for each possible value of $\boldsymbol{Y}$ as $f^*(\boldsymbol{y}, z) = \sum_{\boldsymbol{y}'} \mathbb{1}\{\boldsymbol{y} = \boldsymbol{y}'\} f^*(\boldsymbol{y}', z)$ and write the indicator as products of $y_i$ and $1 - y_i$. Then use the linearity of the expected value as well as the independence of $\boldsymbol{M}$. For details, see supplementary B.3. □

Unfortunately, the sum over $\boldsymbol{y}' \preceq \boldsymbol{y}$ makes computation of Equation 53 computationally expensive. However, in cases with large label spaces, the actual (observed) label vectors typically become quite sparse (Busa-Fekete et al., 2022), $\|\boldsymbol{y}\|_1 \ll m$, which mitigates the impact to some degree: If we assume the number of observed positives to follow $\mathcal{O}(\log(l))$, then we would need

$$2^{\mathcal{O}(\log(l))} = \mathcal{O}(l) \tag{54}$$

evaluations of the original loss function. However, because of the exponential dependency, much more important than the average number of labels is the *maximum* numbe of labels ever encountered in the dataset. For example, in the Wiki-500k dataset, the mean number of positive labels is only 4.7, but in the largest instance that are 274 positives, leading to $2^{274}$ terms that would need to be calculated. If one discarded all points with more than 12 positive labels (5.2% of the dataset), the cost would decrease to only 156 loss evaluations per point – still expensive, but no longer astronomical. Similar statistics for other datasets are presented in Table 1.

Note that this is still much more efficient than the general solution given in Van Rooyen and Williamson (2017, Theorem 5), which is stated in terms of transitions between observations. In our case, each of the $2^m$ possible combination of relevant labels corresponds to one distinct observation. Their unbiased estimate is

$$\mathbf{f}(z) = R^* \mathbf{f}^*(z), \tag{55}$$

where $R^*$ denotes the adjoint of the inverse of the corruption matrix, and $\mathbf{f}$ and $\mathbf{f}^*$ denote vectors that contain the loss for every possible observation. This means that naively, one would need to evaluate all $2^m$ possible values of $f^*(\cdot, z)$ and combine them using a (sparse) $2^m \times 2^m$ matrix.

As in the binary case, we can state a uniqueness theorem:

**Theorem 20** (Multilabel Uniqueness). *Then debiasing operator $\mathfrak{p}_{\boldsymbol{p}}$ from Theorem 19 is unique, i.e., for any $f^* \colon \mathcal{Y}^m \times \mathcal{Z} \longrightarrow \mathbb{R}$, any unbiased version $f$ is equal to $\mathfrak{p}_{\boldsymbol{p}}(f^*)$. Formally, if for all $(Z, \boldsymbol{Y}^*, \boldsymbol{Y}) \in \mathcal{P}_{\boldsymbol{p}}^{\perp\!\!\!\perp}(\mathcal{Z})$ it holds that*

$$\mathbb{E}[f^*(\boldsymbol{Y}^*, Z)] = \mathbb{E}[f(\boldsymbol{Y}, Z)], \tag{56}$$

*then $f = \mathfrak{p}_{\boldsymbol{p}}(f^*)$.*

*Proof.* Since $f$ needs to be unbiased for all possible distributions of $Z$ and $\boldsymbol{Y}^*$ compatible with $\boldsymbol{p}$, it needs to work in particular also for $\mathbb{P}\{Z = z, \boldsymbol{Y}^* = \boldsymbol{y}^*\} = 1$. Since $\boldsymbol{Y}$ can take only finitely many states, we can decompose $f$ into a sum over these states. The claim can then be shown by induction over the number of nonzero elements in $\boldsymbol{y}$, which always introduces only a single new summand in the decomposition. Details can be found in the supplementary B.3. $\qquad\square$

**Lemma 21** (Linearity). *The propensity scoring operator $\mathfrak{p}_{\boldsymbol{p}}$ of Theorem 19 is linear,*

$$\mathfrak{p}_{\boldsymbol{p}}(f^* + g^*) = \mathfrak{p}_{\boldsymbol{p}}(f^*) + \mathfrak{p}_{\boldsymbol{p}}(g^*), \tag{57}$$

*and commutes with functions of the auxillary variable, i.e., for $g^* \colon \mathcal{Z} \longrightarrow \mathbb{R}$ it holds*

$$\mathfrak{p}_{\boldsymbol{p}}(f^* \cdot g^*) = g^* \cdot \mathfrak{p}_{\boldsymbol{p}}(f^*). \tag{58}$$

*Proof.* Both statements can be read directly from Equation 53, by using that the sum is linear and $g^*(z)$ does not depend on the summation variable. $\qquad\square$

## 5.2 Normalized Multilabel Reductions

In Section 3.5, we introduced multilabel reductions that allow training classifiers optimally for precision-at-$k$. With Theorem 19, we can now cover *normalized* reductions, which allow training consistent with recall-at-$k$. We first introduce a general form of a normalized reduction, which allows unified treatment of PAL-N and OVA-N reductions. The normalized reductions are derived from the basic reductions, Equations (34) and (34), by replacing the label $\boldsymbol{Y}^*$ with a normalized label variable $\boldsymbol{T}^*$,

$$\forall k \in [m] \colon \ T_k^* \coloneqq \frac{Y_k^*}{\sum_{j=1}^m Y_j^*} = \frac{Y_k^*}{1 + \sum_{j \neq k}^m Y_j^*}, \tag{59}$$

where the second equality is well defined for all possible label vectors, and equal to the first under the convention $0/0 \coloneqq 0$.

**Remark 22.** *We could also possibly interpret this the following way: Under the assumption that in the original training data, there is almost surely at least one label present ($\mathbb{P}[\|\boldsymbol{Y}^*\|_1 = 0] = 0$), we can set the function to an arbitrary value in these cases, without changing its expectation. However, as labels go missing we will not have $\mathbb{P}[\|\boldsymbol{Y}\|_1 = 0] = 0$, and the case $\boldsymbol{y}' = \boldsymbol{0}$ certainly appears in the summation of Equation 53, which means that for any given observed label vector $\boldsymbol{y}$, the value of the unbiased version of the function will depend on the choice of $0/0$. This might seem to contradict the uniqueness result state above, but note that by assuming $\mathbb{P}[\|\boldsymbol{Y}^*\|_1 = 0] = 0$, we no longer want our unbiased version to work for all possible data distributions.*

Regardless of the underlying interpretation, this leads to the following general form:

**Definition 23** (Normalized Reduction). *A multilabel function $f^*$ is considered to be a* normalized reduction *if there exist functions $g_i^*$ and $h^*$ such that*

$$f^*(\boldsymbol{y}, z) = h^*(z) + \sum_{k=1}^m \frac{y_k}{1 + \sum_{j \neq k} y_j} g_k^*(z). \tag{60}$$

In terms of the normalized labels, we can rewrite the value of such a function as $f^*(\boldsymbol{Y}^*, Z) = \sum_{j=1}^{m} T_j^* g_j^*(Z) + h^*(Z)$. Thus, due to Lemma 21, for calculating the unbiased version it suffices to apply the generic formula of Theorem 19 to $T_i^*$. We get

$$\mathfrak{p}_{\boldsymbol{p}}(T_k^*) =: T_k = \sum_{\boldsymbol{y}' \preceq \boldsymbol{y}} \left( \prod_{j:y_j=1} \frac{y_j'(2-p_j) + p_j - 1}{p_j} \right) \frac{y_k'}{1 + \sum_{j \neq k}^{m} y_j'} \, . \tag{61}$$

By adjusting the summation bounds, we can rewrite this to

$$T_k = \sum_{\boldsymbol{0} \prec \boldsymbol{y}' \preceq \boldsymbol{y}} \left( \prod_{j:y_j=1} \frac{y_j'(2-p_j) + p_j - 1}{p_j} \right) \frac{y_k'}{\|\boldsymbol{y}'\|_1} \, . \tag{62}$$

Plugging this into Lemma 21 and rearranging terms, we arrive at:

**Corollary 24.** *Let $f^*$ be a normalized-reduction, i.e., of the form Equation 60, then its unbiased version $f := \mathfrak{p}_{\boldsymbol{p}}(f^*)$ is given by*

$$f(\boldsymbol{y}, z) = h^*(z) + \sum_{\boldsymbol{0} \prec \boldsymbol{y}' \preceq \boldsymbol{y}} \left( \prod_{j:y_j=1} \frac{y_j'(2-p_j) + p_j - 1}{p_j} \right) \frac{\sum_{k=1}^{m} y_k' g_k^*(z)}{\|\boldsymbol{y}'\|_1} \, . \tag{63}$$

The next paragraphs present the decomposition of two classes of loss functions in terms of Definition 23, so that this corollary can be applied.

**One-vs-All-Normalized**   Let $g_{\mathrm{BC}}$ be a binary loss function.[3] Then the OvA-N reduction is defined as

$$f_{\mathrm{OvA-N}}^*(\boldsymbol{y}, \hat{\boldsymbol{y}}) := \sum_{i=1}^{m} \frac{y_i}{\sum_{j=1}^{m} y_j} g_{\mathrm{BC}}(1, \hat{y}_i) + \left( 1 - \frac{y_i}{\sum_{j=1}^{m} y_j} \right) g_{\mathrm{BC}}(0, \hat{y}_i). \tag{64}$$

By comparing terms, we see that this corresponds to $g_i(\hat{\boldsymbol{y}}) = g_{\mathrm{BC}}(1, \hat{y}_i) - g_{\mathrm{BC}}(0, \hat{y}_i)$ and $h(\hat{\boldsymbol{y}}) = \sum_{i=1}^{m} g_{\mathrm{BC}}(0, \hat{y}_i)$.

**Pick-All-Labels-Normalized**   For a given multiclass loss $g_{\mathrm{MC}}$, the PAL-N reduction is given by

$$f^*(\boldsymbol{y}, \hat{\boldsymbol{y}}) := \sum_{i=1}^{m} \frac{y_i}{\sum_{j=1}^{m} y_j} g_{\mathrm{MC}}(i, \hat{\boldsymbol{y}}), \tag{65}$$

which immediately gives the correspondence $g_i(\hat{\boldsymbol{y}}) = g_{\mathrm{MC}}(i, \hat{\boldsymbol{y}})$ and $h \equiv 0$.

**Instance-Averaged Recall**   In a multilabel setting, we can define the instance-averaged recall[4] as the fraction of relevant labels that have been predicted (Lapin et al., 2018). Thus for $\hat{\boldsymbol{y}} \in \{0,1\}^l$

$$\mathrm{Rec}^*(\boldsymbol{y}, \hat{\boldsymbol{y}}) := \|\boldsymbol{y}\|_1^{-1} \sum_{j=1}^{m} y_j \hat{y}_j. \tag{66}$$

This can be interpreted both as a normalized PAL reduction with $g_{\mathrm{MC}}(i, \hat{\boldsymbol{y}}) = \hat{y}_i$, and as normalized OvA with $g_{\mathrm{BC}}(y, \hat{y}) = y\hat{y}$. In Appendix A we provide an explicit formula for the unbiased estimator.

---

[3] Generalization to label-dependent loss functions is straightforward.
[4] opposed to micro-, or macro-averaging

## 6 Upper-Bounds

The unbiased version allows us to calculate the loss even on data with missing labels, but can we also use it for training? Ideally, the loss function should be lower-bounded, so the minimization is well defined, it should be convex so the minimum is unique. Further, the variance of the unbiased estimate should not be too large, so that a reasonable amount of training samples is sufficient.

If we assume $\ell_{\mathrm{BC}}$ and $\ell_{\mathrm{MC}}$ to be lower-bounded and convex, then only the PAL-reduction results in an unbiased estimate that is guaranteed to have the same properties, as it is a positive combination of $\ell_{\mathrm{MC}}$. Due to the uniqueness result, it is not possible to find an unbiased version that is always convex for the other reductions. Thus, in order to make them amenable for training, we propose to switch from unbiased estimates to convex upper-bounds.

### 6.1 Convex Upper Bounds for OvA Losses

Many binary losses used in machine learning are convex surrogates of the 0-1-loss, for example the logistic loss, (squared) hinge loss, and squared error. Thus, one way to arrive at convex loss functions adapted to missing labels is to switch the order of operations(Qaraei et al., 2021; Chou et al., 2020): Instead of calculating unbiased versions of convex surrogates, we calculate the unbiased version of the 0-1-loss and take a convex surrogate of the resulting expression.

To that end, first consider the binary case: Let $\ell^* \colon \{0,1\} \times \mathbb{R} \longrightarrow \mathbb{R}$ be a function that is convex in its second argument and forms an upper-bound on the 0-1 loss. The unbiased version for the 0-1 loss for positive label is given by

$$
\begin{aligned}
\mathfrak{P}_p(\ell_{01})(1,\hat{y}) &= p^{-1}\left(\mathbb{1}\{\hat{y} \leq 0\} + (p-1)\mathbb{1}\{\hat{y} > 0\}\right) \\
&= p^{-1}\left(\mathbb{1}\{\hat{y} \leq 0\} + (p-1)(1 - \mathbb{1}\{\hat{y} \leq 0\})\right) \\
&= p^{-1}\left((2-p)\mathbb{1}\{\hat{y} \leq 0\} + p - 1)\right) \\
&= (2/p - 1)\,\mathbb{1}\{\hat{y} \leq 0\} + \mathrm{const.}
\end{aligned}
\tag{67}
$$

From an empirical-risk minimization perspective, the constant does not affect the outcome, so it can be ignored. Therefore, a convex upper-bound to this unbiased 0-1 loss (with the constant removed) is given by

$$
f(y,\hat{y}) = y\left(\frac{2}{p} - 1\right) f^*(1,\hat{y}) + (1-y)f^*(0,\hat{y}).
\tag{68}
$$

This type of function has been shown in Qaraei et al. (2021) to improve performance on several XMC datasets.

### 6.2 Upper Bounds for Normalized Multilabel Reductions

A naive attempt of correcting for the noisy labels by replacing $\boldsymbol{Y}^*$ with $Y/p$ in the calculations for the normalized labels $\boldsymbol{T}^*$. However, the resulting estimator $\tilde{\boldsymbol{T}}$ turns out to be an upper bound. The two estimators are given by

$$
T_i^* = \frac{Y_i^*}{1 + \sum_{j \neq i} Y_j^*}, \qquad \tilde{T}_i := \frac{Y[i]/p_i}{1 + \sum_{j \neq i} Y[j]/p_j}.
\tag{69}
$$

**Theorem 25** (Normalized Label Upper-Bound). *Let $(Z, \boldsymbol{Y}^*, \boldsymbol{Y}) \in \mathcal{P}_{\boldsymbol{p}}^{\perp\!\!\!\perp}(\mathcal{Z})$, replacing the true labels with the unbiased estimate of the observed labels as shown in Equation 69 results in an upper bound, whose error itself can be bounded by a data-dependent term*

$$
\mathbb{E}[T_i^*] + \sum_{j \neq i}\left(\frac{1 - p_j}{p_j}\right)\mathbb{E}\left[\frac{Y_i}{p_i} \cdot \frac{Y_j}{p_j}\right] \geq \mathbb{E}\left[\tilde{T}_i\right] \geq \mathbb{E}[T_i^*].
\tag{70}
$$

The proof is presented in subsection B.4 and relies on the transformation of equation (59), as it makes the mask variables in the numerator and denominator independent.

In practice, most entries of the co-occurrence matrix $\mathbb{E}[Y_i \cdot Y_j]$ will be extremely small, causing only a minute contribution to the error bound. This can be illustrated by calculating, on two real datasets, the upper-bound for the error of the proposed estimator, by approximating $\mathbb{E}[Y_i \cdot Y_j]$ with the empirical label co-occurrence frequency. The propensities are estimated as in Jain et al. (2016). Looking at the mean value, and the worst case for any label (Table 2), We can see that the error on average is very small, indicating that the worst-case bound only applies to very few labels.

Table 2: Error bound for XMC datasets

| Dataset | Average | Worst Case |
|---|---|---|
| Eurlex-4K | 0.02 | 0.51 |
| AmazonCat-13K | 0.0006 | 0.24 |

**Corollary 26** (PAL Upper-Bound). *Under the assumptions of Theorem 25, if the underlying multiclass loss $\ell_{\mathrm{MC}}$ is a non-negative convex function, the expression*

$$\tilde{\ell}(\mathbf{y}, \hat{\mathbf{y}}) := \sum_{j=1}^{m} \frac{y_i/p_i}{1 + \sum_{j \neq i} y_j/p_j} \ell_{\mathrm{MC}}(j, \hat{\mathbf{y}}) \tag{71}$$

*gives a nonnegative, convex upper-bound on the true normalized PAL loss in expectation.*

## 7 Evaluation Experiments

In this section, we validate the result of Theorem 19 experimentally using the example of recall. We first provide a look at synthetic data, where propensities are exactly known and we can calculate reference ground-truth values. This will show that even in an ideal setting, if the propensities get too low, the variance of the estimate increases so much that it can become unusable. Then we apply the estimator to real predictions, demonstrating that it is applicable for some datasets, but useless for others.

Consider a setting in which there are 100 different labels, which are independent and each has a probability of 10%. We randomly draw 10 000 ground-truth label vectors, and generate observed labels by removing according to a propensity $p$ that is identical for all labels. The predictions are generated by randomly choosing a label from the ground-truth. We calculate the average per-example recall using the vanilla estimator, the unbiased estimator, and the upper bound, and plot the results in Figure 2.

The figure shows that for propensities lower than 0.4, the unbiased estimate's variance becomes too large. The upper bound also becomes unsuitable, because it deviates strongly from the actual value, and the (biased) vanilla estimate ends up closest to the true value. For medium propensity, the unbiased estimator still has little variance, but noticeable better accuracy than the vanilla one. Together with the uniqueness Theorem 20, these results suggest that for datasets with very low propensity, unbiased estimates are ill-suited to calculate the per-example recall.

Looking at the results presented in Bhatia et al. (2016), one notices that recall metrics are conspicuously absent. This is presumably because they are not straightforward to compute. In Jain et al. (2016) there is a

Table 3: Propensity-scored (77) and vanilla (66) recall at $k$ for DiSMEC-style models trained using the squared hinge convex surrogate Qaraei et al. (2021). The Filter column indicates the fraction of data points that had to be removed as outliers to decrease the variance to reasonable levels.

| Dataset/k | PsRec@k | | | Rec@k | | | Filter |
|---|---|---|---|---|---|---|---|
| | 1 | 3 | 5 | 1 | 3 | 5 | |
| Bibtex | 42.1 | 65.9 | 78.3 | 33.6 | 53.0 | 62.0 | 0% |
| Mediamill | 21.6 | 49.2 | 61.9 | 23.3 | 52.6 | 65.6 | 0% |
| RCV1-2K | 42.4 | 78.1 | 82.3 | 40.4 | 74.4 | 80.7 | 1% |
| AmazonCat-13K | 24.2 | 57.1 | 72.3 | 26.4 | 59.7 | 75.0 | 0.05% |
| AmazonCat-14K | 38.8 | 68.1 | 83.0 | 39.4 | 69.3 | 83.9 | 0.01% |

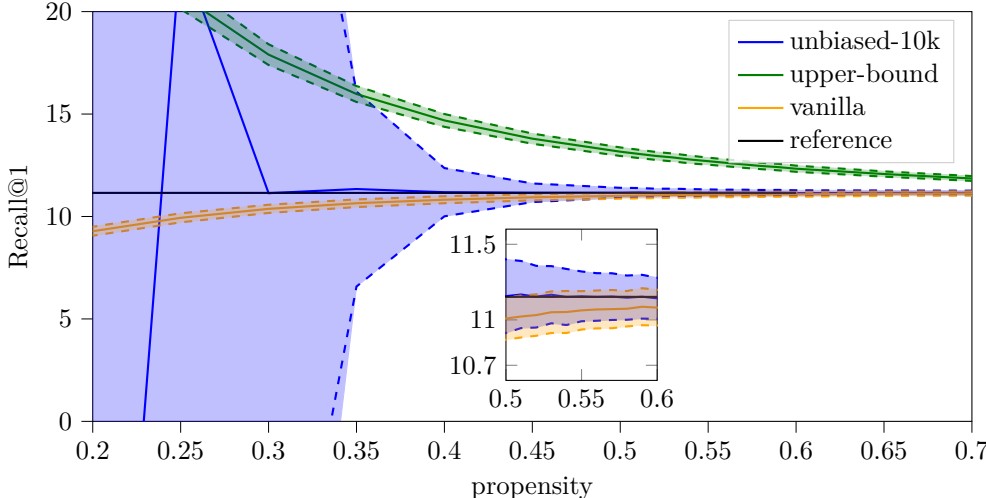

Figure 2: Unbiased estimate of per-example recall with artificial data as described in the main text. The shaded region corresponds to one standard deviation, estimated over 100 repetitions. The black line denotes the true recall.

section that argues how to calculate this in cases where the total number of labels is available, but for other cases this paper leaves a gap, which is filled by Theorem 19. Unfortunately, as the results with synthetic data suggest, once the propensities become too low, the method becomes unusable. This precludes its application to datasets like `Amazon-670K` or `EURLex-4K`.

To generate the predictions, we trained a DiSMEC (Babbar and Schölkopf, 2017) model using a convex surrogate unbiased loss function (Qaraei et al., 2021) for the squared-hinge loss. Two difficulties arise when when calculating the unbiased recall estimate: First, even if the average number of relevant labels is low, there still can be some samples with a high number of true labels, which becomes prohibitively expensive in computation. This can be handled by a sampling approach. The second problem is that the unbiased estimate comes at the cost of vastly increased variance. In fact, we observed that the mean is dominated by very few outlier samples, and obtained nonsensical values. Therefore, we filtered out values in the lowest and highest quantiles. This results in a biased estimate, but gives much reduced variance. Details can be found in supplementary C.1.

Table 3 shows that the recall values for vanilla and unbiased estimation are close except for the Bibtex dataset. A possible reason could be that the propensity model, which was empirically found for very large datasets, does not fit appropriately for this small dataset. For other datasets on the repository (Bhatia et al., 2016), `Amazon-670K`, `WikiLSHTC-325K` and `EURLex-4K`, the variance in the unbiased estimate is too large to get a meaningful result.

# 8   Training Experiments

Ideally, we would benchmark our loss functions on a task based on real XMC datasets. However, for those these we neither know the exact propensities, nor can we validate that the unbiased estimates and upper bounds produce reasonable results, since the fully-labeled ground truth is unknown. For example, Schultheis et al. (2022, Table 3) shows that even with the particularly benign unbiased estimator for propenstiy-scored precision, the quality of available propensity estimates is too low to allow for meaningful results. Consequently, we turn to more carefully constructed data to analyse our proposed loss functions.

## 8.1  Experimental Setup

Instead of using fully artificial data, we chose to construct a dataset based on existing realistic data: We took the `AmazonCat-13k` data and consider only the 100 most common labels, which are the ones with the highest propensity according to the Jain et al. (2016) model. We artificially remove labels according to inverse propensity, which increases linearly based on the ordering of label frequencies, such that the most common label has an inverse propensity of two and the 100th most common one has an inverse propensity of 20. This process partially preserves the strong imbalances that are typical of extreme classification datasets.

On this data, we train a linear classifier with $L_2$-regularization using different basis loss functions with **a)** the original loss on clean training data and **b)** noisy training data, as well as **c)** the unbiased version and **d)** the upper-bound version on noisy data. For each training run, we evaluate training loss, as well as the task losses precision and recall at k, on noisy and clean training and test data. For the evaluation on noisy data, the corresponding unbiased estimators are used.

By training with the vanilla loss, we get an upper bound how well the given network could do without noise (config **a**) and how poorly it would perform without any mitigations (config **b**). In the first setting, we expect the calculated loss on clean and noisy data to match, since the model cannot overfit to the specific noise pattern in the training data, whereas in configurations **b)** to **d)** we expect to see a difference. As we remove labels independently of features, the unbiased estimate on noisy test data should be equal to the actual value on clean test data, if the test set contains sufficient samples to result in accurate estimates.

We took 30% of the original training data and used them as validation data to determine the optimal value for the strength of $L_2$-regularization. As the choice of loss function influences the bias-variance trade-off, this value needs to be determined for each configuration. Note that when we train with noisy data, we also use noisy data for validation, i.e. we assume a setting in which no clean data is available at all.

The network is optimized using Adam (Kingma and Ba, 2017) with an initial learning rate of $10^{-4}$ for the first 15 epochs and $10^{-5}$ for the remaining five epochs, with a mini-batch size of 512.

## 8.2  Overfitting to Sample and Noise Pattern

In addition to the overfitting that is due to the finiteness of the training data, the missing-labels setting causes additional overfitting because only a single realization of the noise pattern is observed.

Let $h$ be a classifier that depends on the observed data $Y$, and $f^*$ a loss function with estimator $f = \mathfrak{P}_p(f^*)$. Then the generalization decomposes into

$$\mathrm{R}_{f*}^*[h] - \hat{\mathrm{R}}_f[h] = \underbrace{\mathrm{R}_{f*}^*[h] - \hat{\mathrm{R}}_{f*}^*[h]}_{\text{finite sample}} + \underbrace{\hat{\mathrm{R}}_{f*}^*[h] - \hat{\mathrm{R}}_f[h]}_{\text{noise pattern}}, \tag{72}$$

the difference between the true risk $\mathrm{R}_{f*}^*[h]$ and the empirical risk on clean training data $\hat{\mathrm{R}}_{f*}^*[h]$, and the difference between that and the estimated empirical risk on observed data $\hat{\mathrm{R}}_f[h]$. Because the classifier $h$ depends (through $Y = \mathbf{M} \odot Y^*$) on the mask variables, $f$ does not give an unbiased estimate (on training data) and thus the second term is non-zero even in expectation.

In fact, in the linear classifier experiment described here, the noise-pattern overfitting is much stronger than the overfitting due to finite sampling. Figure 3 shows this for the case of the BCE loss in OvA-reduction, though the same effect can be seen also for the other loss functions, see supplementary. For the classifier trained on clean data (blue), the weights are independent of the noise pattern and thus the dashed and dotted lines coincide in expectation.

We can see that the label noise acts as an implicit regularizer, in the sense that for low regularization, the gap between clean test and clean training data is much lower when the classifier is trained on noisy data as compared to clean training data.

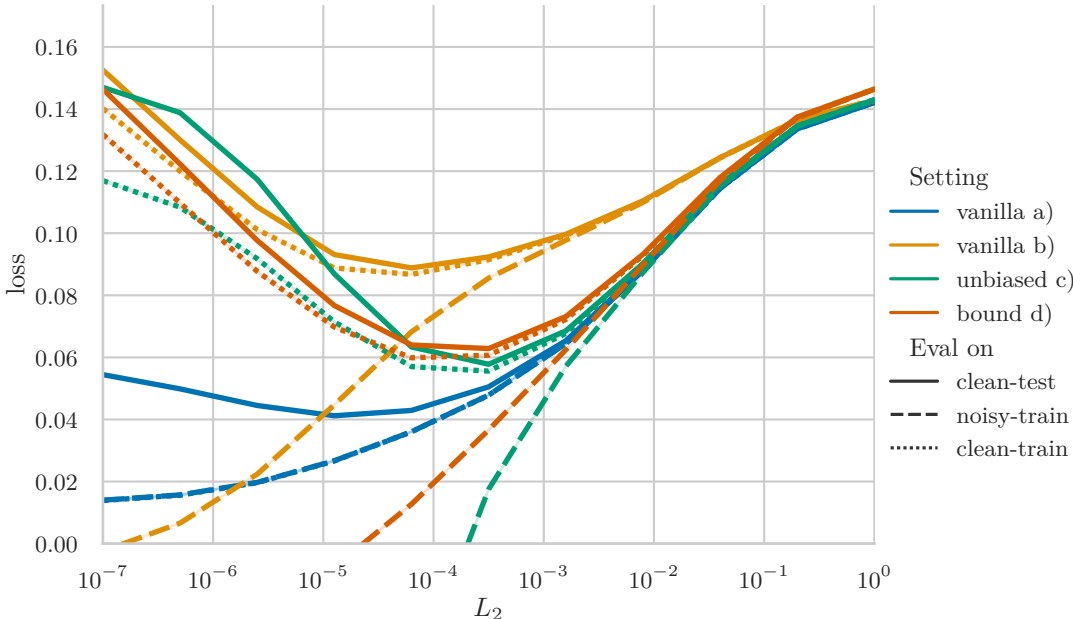

Figure 3: Binary cross-entropy for different regularization strengths, evaluated on noisy training data, clean training data, and clean test data. The gaps between dashed and dotted lines correspond to overfitting to the noise pattern, the smaller gaps between dotted and solid lines show the generalization gaps due to the finite training sample. As the dashed lines are for noisy data, they are calculated using the unbiased estimate (34).

### 8.3 Results for Loss Minimization

As Figure 3 shows, for the case of OvA reduction using the BCE loss, the training loss gets reduced much further using the unbiased loss function or the upper-bound loss function than using the vanilla loss. This decrease more than compensates the increase in generalization gap, and as such the minimal test loss, i.e. the loss at optimal regularization, is better with these two variants of the loss function.

Does this improvement also occur for other reductions and loss functions? This question is addressed by Figure 4, which plots the loss on clean test data over varying $L_2$ regularization strength for different base losses.

The first row of graphs shows BCE-based losses, with the left graph depicting the same data as Figure 3. The right graph is for the normalized reduction, where the increased overfitting exceeds the improvement in training loss, and overall the unbiased training produces worse results than just training with the vanilla loss. We have also included an evaluation using the upper-bound formula (69). Note that for the normalized BCE loss, that formula does not result in an upper bound, but the graph shows that it empirically works better than the unbiased loss, yet still worse than vanilla.

A surprising feature of the normalized BCE result is that even for very strong regularization, the unbiased estimate underperforms vanilla loss. We attribute this to the non-convexity of (64) because the increase in loss is due to large training loss as opposed to the generalization gap. By choosing different initial weights (see supplementary C.2), we can reduce the training loss for unbiased training to that of vanilla training, supporting the hypothesis that sub-optimal minima are the problem. Compared to the decomposable case (which is also non-convex), the normalized case requires scaling by products of several inverse propensities, and thus induces much larger prefactors in front of the terms that cause the non-convexity, which might explain why the same phenomenon does not occur in the decomposable reduction.

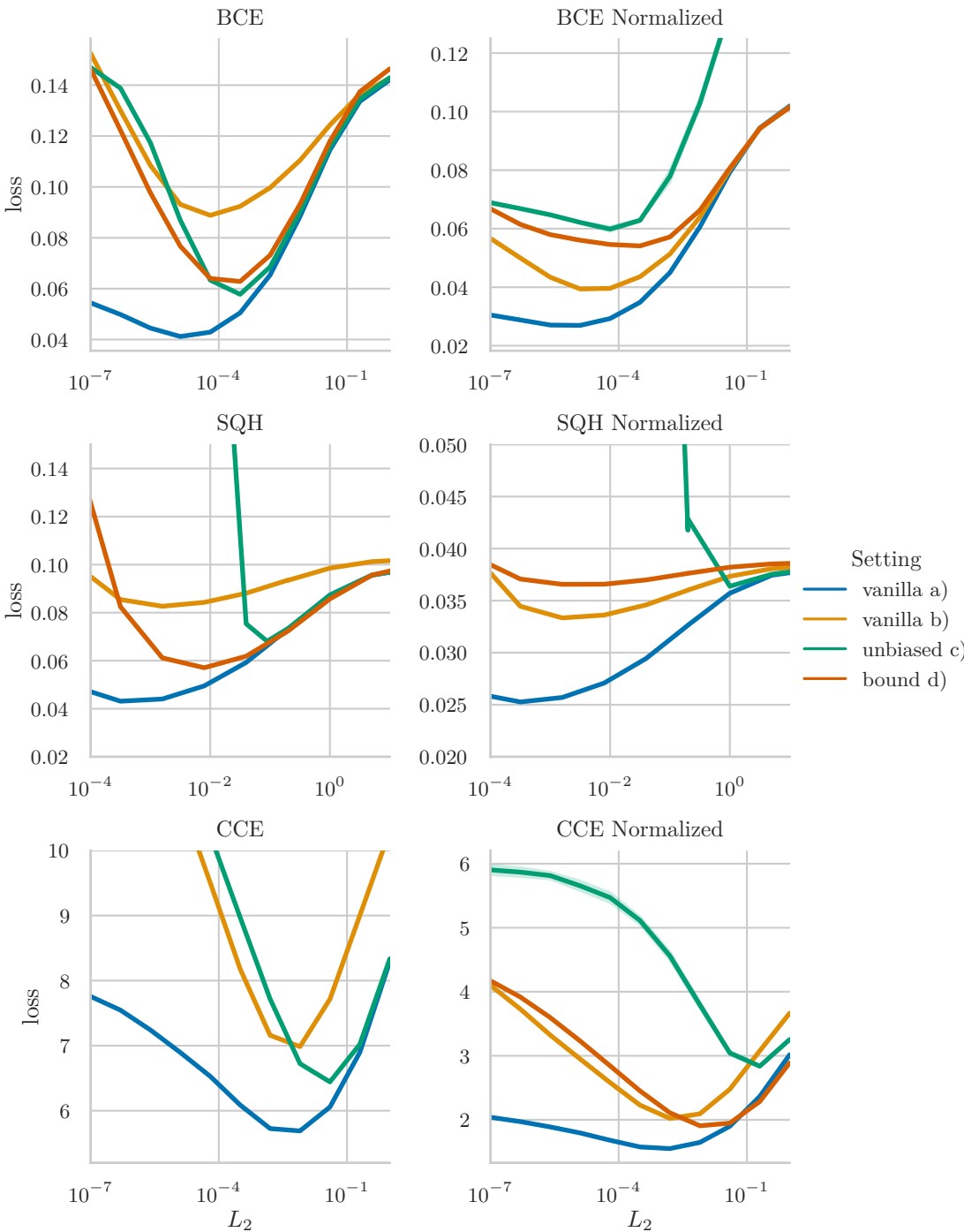

Figure 4: Comparison of different loss functions, evaluated on clean test data, in combination with different schemes for addressing missing labels. Details are in the main text.

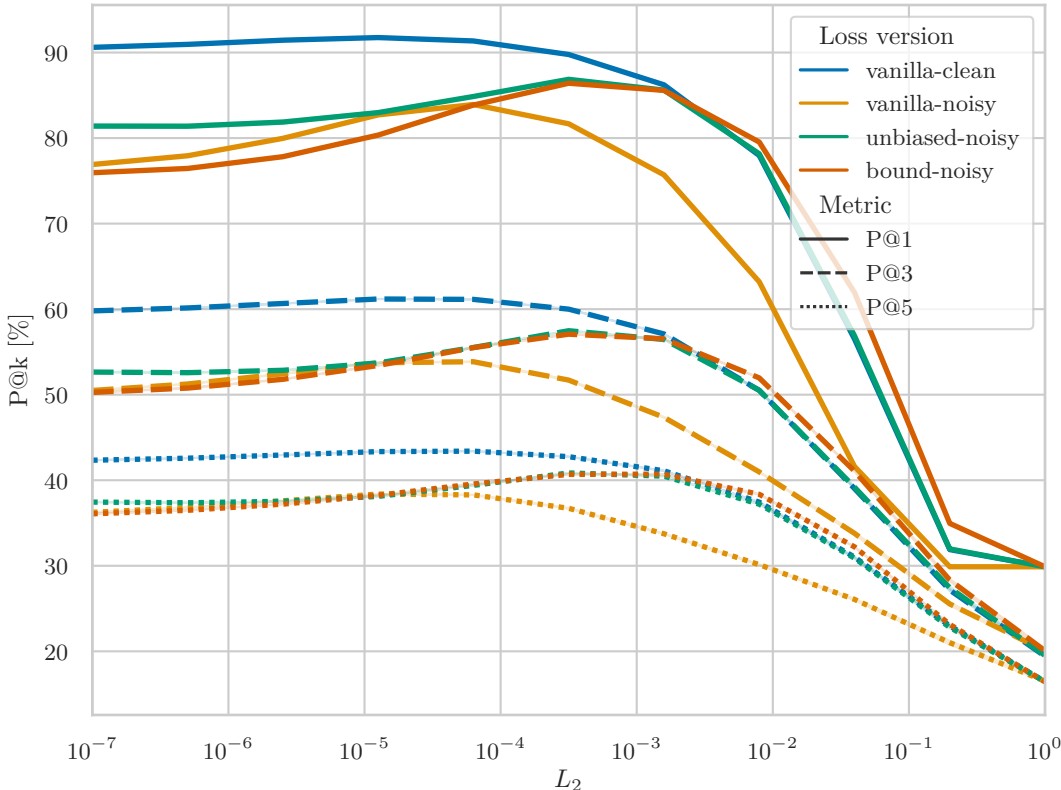

Figure 5: Precision at $k$ for training with binary cross-entropy (normalized reduction) for the different loss variations.

The second row shows squared-hinge based losses, which are also subject to the One-vs-All reduction. The main difference between this loss and BCE is that squared-hinge is even more susceptible to overfitting in the unbiased case. Whereas for BCE the unbiased and upper-bound variations give roughly comparable results, in this case the upper-bound loss proved to be much more stable. In the normalized setting, the unbiased loss becomes very large for even larger regularization strength ($\approx 10^{-1}$) compared to the decomposable variant ($\approx 10^{-2}$). As in the BCE-normalized setting, the expression in (69) is not actually an upper bound, and we can see empirically that it does not work well here. We again can observe that the vanilla loss performs better than the unbiased loss.

The third row shows categorical cross-entropy, which differs from the other two rows in that it results in a Pick-all-Labels reduction. In the non-normalized case, the upper bound and unbiased loss are identical and outperform the vanilla loss. The normalized case shows the same problems as above for the unbiased loss, but the upper bound (71) successfully improves on the vanilla loss.

In terms of the bias-variance trade-off, the graphs show a clear trend: The optimal regularization for training on noisy data is larger than on clean data. It is also larger when using the unbiased or upper-bound loss as compared to vanilla loss.

### 8.4  Task Losses

Even though the training process on clean data is based on the optimization of a decomposition into a differentiable loss, this is typically not the quantity that is ultimately of interest. Instead, what we really want is a maximization of precision or recall at the top (Kar et al., 2015).

The behaviour of precision at k in dependence of the regularization is depicted in Figure 5 for the BCE loss. There are some notable differences in the behaviour of the P@k metrics compared to the loss function: Whereas the increased overfitting for low regularization results in a strong deterioration in the loss function, the corresponding decrease in precision metrics is relatively mild. For the loss function, the unbiased estimate works better at higher regularization and worse at lower regularization than the upper bound, but here the situation is reversed.

The full results for precision and recall at the optimal regularization parameter are presented in Table 4. In this context, optimal is to be understood as the value for which the unbiased estimate of the loss function on noisy validation data is minimized. These values might be slightly different than the optimal values for a specific P@k or R@k metric. For a visual representation of this data, see subsection C.3.

Comparing the decomposable and the normalized settings, we can see that the normalized losses induce larger fluctuations (measured in standard deviation) for the OvA settings and the unbiased PAL setting, with the exception of vanilla training on noisy data b) for BCE loss. This holds across all six task losses. Note that this cannot be explained by the variance in the unbiased estimators, as the values presented here have been calculated directly on the clean ground-truth test data.

The graph also shows that the ordering of settings a)-d) is mostly stable (for fixed loss function) over the six different metrics, but differs when switching loss function. For OvA-BCE, the unbiased and upper-bound losses c) and d) are almost equal, but for OvA-SQH the unbiased performs far worse, and with much larger fluctuations, which is consistent with the observations in Figure 4. For PAL-CCE, the two settings are the same.

In the normalized cases, we can see that the OvA-N reduction results in mostly the same ordering for both BCE and SQH: The vanilla loss b) is better than applying (71) d), which in turn is better than the unbiased loss c). The exception are the recall metrics for BCE, where the unbiased loss appears to give better results.

For the PAL-N reduction we can see that the upper bound d) does result in better task loss than using vanilla loss b), but as with the other normalized setups the unbiased loss c) performs worse.

Finally, we want to know in which circumstances the normalized variations perform better than the decomposable ones. The results in Menon et al. (2019) prove that in the asymptotic case the normalized reductions are consistent for recall whereas the others are consistent for precision. However, as seen above, despite the large number of training instances in the dataset, overfitting is still a major problem, and thus it may well be that a method that is consistent for recall also gets better results in precision if its generalization gap is significantly smaller.

From Table 4 we can see that for the BCE loss the normalized reduction generally results in worse performance for both precision and recall, with the notable exception of using vanilla loss on noisy data, where there is a slight increase in recall. Nonetheless, using unbiased or upper-bound losses for the decomposable reduction results in overall better performance. For the squared hinge (SQH) loss, the normalized reduction performs worse for all variations and all metrics.

For the decomposable CCE reduction, the upper bound is equal to the unbiased loss. Compared with the bound for the normalized reduction, the latter gives better values in P@1 and R@1, but worse for the other metrics. The unbiased normalized reduction performs much worse across all metrics. For the vanilla loss, the results are as expected according to Menon et al. (2019) with the normalized version yielding better recall. Interestingly, when used on noisy data, it also gives better precision.

Table 4: Training results on modified AmazonCat-13K data for using different loss functions in their **V**anilla, **U**nbiased or Upper-**B**ounded variants. BCE denotes the binary cross-entropy and SQH the squared-hinge loss corresponding to a OvA decomposition, and CCE denotes (softmax) categorical cross-entropy corresponding to a PaL decomposition. The settings marked with a* denote reference runs on clean data.

| Setting | Precision | | | Recall | | | Loss | Reg. |
|---------|------|------|------|------|------|------|------|------|
| | P@1 | P@3 | P@5 | R@1 | R@3 | R@5 | | |
| B-BCE | 86.4 | 57.1 | 40.7 | 46.1 | 75.9 | 84.4 | $6.3 \cdot 10^{-2}$ | $3.16 \cdot 10^{-4}$ |
| B-NBCE | 75.8 | 45.9 | 32.1 | 37.1 | 60.0 | 67.5 | $5.6 \cdot 10^{-2}$ | $3.4 \cdot 10^{-4}$ |
| U-BCE | 86.9 | 57.5 | 40.9 | 46.2 | 76.1 | 84.5 | $5.8 \cdot 10^{-2}$ | $3.16 \cdot 10^{-4}$ |
| U-NBCE | 72.5 | 45.3 | 32.3 | 40.5 | 64.5 | 72.3 | $6.2 \cdot 10^{-2}$ | $3.41 \cdot 10^{-5}$ |
| V-BCE* | 91.7 | 61.2 | 43.3 | 50.0 | 81.0 | 88.9 | $4.1 \cdot 10^{-2}$ | $1.26 \cdot 10^{-5}$ |
| V-NBCE* | 87.7 | 58.7 | 41.9 | 48.6 | 79.1 | 87.3 | $2.8 \cdot 10^{-2}$ | $4.73 \cdot 10^{-6}$ |
| V-BCE | 83.9 | 53.9 | 38.3 | 43.8 | 71.7 | 80.2 | $8.9 \cdot 10^{-2}$ | $6.31 \cdot 10^{-5}$ |
| V-NBCE | 83.6 | 53.6 | 38.1 | 43.9 | 71.9 | 80.3 | $3.9 \cdot 10^{-2}$ | $2.1 \cdot 10^{-5}$ |
| B-CCE | 86.4 | 56.9 | 40.6 | 46.3 | 76.0 | 84.5 | 6.4 | $3.98 \cdot 10^{-2}$ |
| B-NCCE | 86.7 | 56.3 | 40.0 | 46.7 | 75.9 | 84.1 | 1.9 | $2.39 \cdot 10^{-2}$ |
| U-CCE | 86.4 | 56.9 | 40.6 | 46.3 | 76.0 | 84.5 | 6.4 | $3.98 \cdot 10^{-2}$ |
| U-NCCE | 71.9 | 43.8 | 30.9 | 40.6 | 64.3 | 71.7 | 3.3 | 0.13 |
| V-CCE* | 89.4 | 59.9 | 42.8 | 48.8 | 79.9 | 88.2 | 5.7 | $7.94 \cdot 10^{-3}$ |
| V-NCCE* | 88.9 | 59.8 | 42.6 | 49.0 | 80.2 | 88.5 | 1.6 | $4.43 \cdot 10^{-4}$ |
| V-CCE | 85.0 | 53.9 | 38.3 | 44.8 | 72.2 | 80.5 | 7 | $7.94 \cdot 10^{-3}$ |
| V-NCCE | 85.6 | 54.9 | 39.1 | 45.7 | 74.1 | 82.5 | 2.1 | $2.06 \cdot 10^{-3}$ |
| B-SQH | 86.2 | 56.5 | 40.4 | 45.8 | 75.0 | 83.8 | $5.7 \cdot 10^{-2}$ | $7.94 \cdot 10^{-3}$ |
| B-NSQH | 60.9 | 38.1 | 28.3 | 28.4 | 49.0 | 58.7 | $3.8 \cdot 10^{-2}$ | 0.21 |
| U-SQH | 73.2 | 48.7 | 36.2 | 36.8 | 64.5 | 76.1 | $1.7 \cdot 10^{-1}$ | 0.1 |
| U-NSQH | 49.9 | 32.6 | 24.1 | 26.5 | 47.9 | 56.9 | $1.9 \cdot 10^{-1}$ | 0.43 |
| V-SQH* | 91.7 | 61.0 | 43.1 | 49.9 | 80.7 | 88.5 | $4.3 \cdot 10^{-2}$ | $3.16 \cdot 10^{-4}$ |
| V-NSQH* | 89.2 | 58.1 | 41.2 | 49.4 | 78.8 | 86.5 | $2.6 \cdot 10^{-2}$ | $7.44 \cdot 10^{-4}$ |
| V-SQH | 82.4 | 52.1 | 37.4 | 42.5 | 69.4 | 78.3 | $8.3 \cdot 10^{-2}$ | $1.58 \cdot 10^{-3}$ |
| V-NSQH | 79.7 | 49.7 | 35.8 | 40.7 | 66.6 | 75.9 | $3.4 \cdot 10^{-2}$ | $1.07 \cdot 10^{-2}$ |

## 8.5 Real data: Yahoo-music-R3

To perform an evaluation on actual real-world data, we repeat the experiment above on the YahooMusic-R3 dataset(Yahoo! Inc.). While this is not a traditional XMC dataset, containing ratings that users have given to songs, we can turn it into a multilabel problem as follows: Pick half the available songs and use users' ratings as features. For the other half, consider them to be a positive label if the user's rating is above 4 (out of 5). Crucially, while the training set has been collected like typical XMC datasets, that is, users decided which songs they rated, the test set is based on explicitly querying the user on 10 out of the 1000 possible songs chosen *uniformly* at random.

On the test set, we can estimate the true label prior by taking the ratio of labeled positives to the total number of labeled instances for each song. The ratio of the observed training set prior to the true label prior gives the propensity.

We chose 500 of the songs randomly to be used as features, based on which the rating of th other 500 songs are to be predicted. We remove any labels for which there are no positives, or for which the estimated propensity is below 0.01, which leaves 459 valid labels. To normalize the feature vectors, we apply a tf-idf-like transform, that is, the feature values are scaled by $\log(n/n_{\text{positive}})$.

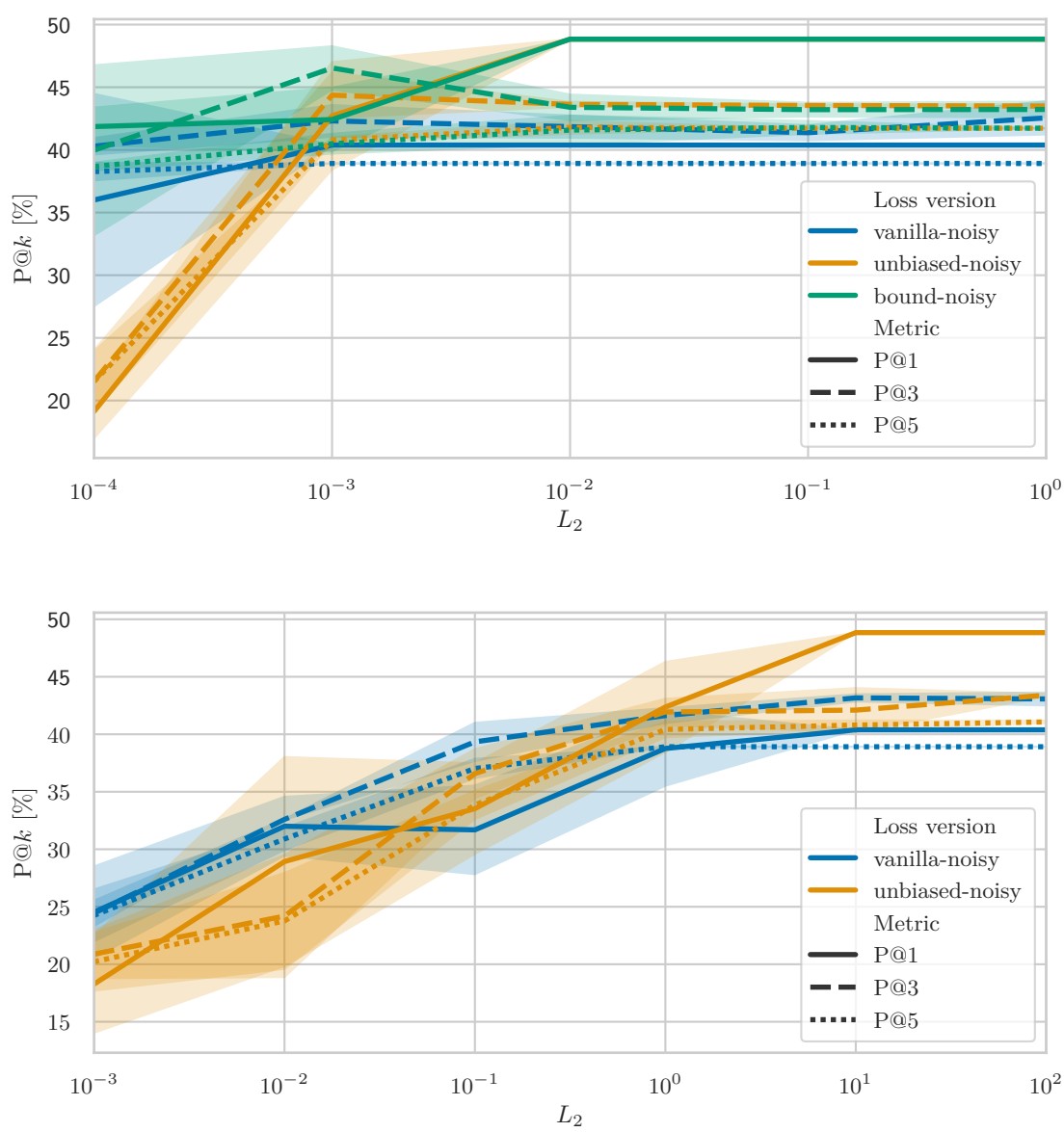

Figure 6: Test set performance of linear classifiers trained on Yahoo-music-R3 with varying levels of regularization, using binary cross-entropy / one-vs-all (top) and categorical cross-entropy / pick-all-labels (bottom).

As above, we train a linear classifier over a sweep of regularization strengths, as shown in Figure 6. The graphs show that, at strong regularization, the unbiased loss outperforms using the original loss function. For BCE, at high regularization the unbiased and upper-bound functions perform equally, but at low regularization, the unbiased loss becomes very bad. In fact, one can see that (with the exception of P@3 for BCE) best performance is achieved at the strongest regularization level, where the resulting PSP@k plateaus, contrary to the expectation that too strong regularization would be detrimental to performance.

The plateau, and the vanishingly small error bars, happen because the classifier degenerates to a constant function, as all the multiplicative weights are driven to zero. In that case, the classifier needs to predict the labels with the highest class prior, and the effect of the unbiased losses essentially boils down to ensureing

the true class prior is estimated, instead of using the observed class prior. Thus, while this example shows that these losses work as intended, the actual learning happening here is rather trivial.

## 9 Related Work

**Unbiased Estimates for Noisy Labels**  Learning with missing labels is a specific instance of learning with class-conditional noise. For the case of binary labels, unbiased estimates of the loss function can be found in Natarajan et al. (2017). Note that the generalization bound given therein (Lemma 8) is missing a factor of max norm of the adapted loss function $\|\tilde{\ell}\|_\infty$, cf. Theorem 9.

An even more general approach is given in Van Rooyen and Williamson (2017). In their notation, $f$ is a function and $\mathbb{P}$ the probability distribution over clean data, that is transformed by the invertible operator $\mathsf{T}$ into a *corrupted* probability distribution. Let $\mathsf{R}$ be the inverse of $\mathsf{T}$, and $\mathsf{R}^*$ its adjoint, then it holds

$$\langle \mathbb{P}, f \rangle = \langle \mathsf{R} \circ \mathsf{T}(\mathbb{P}), f \rangle = \langle \mathsf{T}(\mathbb{P}), \mathsf{R}^*(f) \rangle. \tag{73}$$

This equation forms the basis for their "Theorem 5 (Corruption Corrected Loss)", which states that a *corruption corrected* function $l_\mathsf{R}$ is given by

$$l_\mathsf{R}(\cdot, a) = \mathsf{R}^*(l(\cdot, a)) \; \forall a \in \mathcal{A}, \tag{74}$$

where $\mathcal{A}$ denotes the set of possible actions that will be evaluated by the loss functions. For a finite label space with $n$ possible, the operator $\mathsf{R}^*$ can be represented with an $n \times n$ matrix.

In the specific case of multilabel classification, our results Theorem 19 show that out of the $2^l$ possible values of the label vector that are necessary to evaluate (74) in the general case, in fact only these that are a subset of the observed labels need to be taken into account, thus requiring only $2^{\|y\|_0}$ evaluations.

**Robust Loss-Functions**  An even more robust approach than choosing a loss function which compensates for noisy labels is to use a learning algorithm that is inherently noise tolerant. This has the advantage that one does not need to estimate the noise rate, and cannot introduce additional error by misspecification. For symmetric label noise with rates less than 50%, Ghosh et al. (2015) proved that methods minimizing losses which fulfill a symmetry condition $f(0, \cdot) + f(1, \cdot) = c$ for some constant $c$ are noise tolerant. If a convex loss function is desired, this leads to the *unhinged* loss (Van Rooyen et al., 2015).

Certain performance objectives such as the balanced error or the AUC are noise robust even under the more general setting of mutually contaminated distributions as shown in Menon et al. (2015).

**Data Re-Calibration**  A data re-calibration approach tries to identify from the training data which samples are corrupted (Han et al., 2018; Zheng et al., 2020; Jiang et al., 2018). The co-teaching approach (Han et al., 2018) maintains two interacting networks: For each minibatch, each network selects a subset of examples with low loss value, which is assumed to indicate that these are clean instances. The weights of the other network are then updated by training only on these selected examples. Given that deep neural networks have been observed to initially fit clean data and start overfitting on noise as the training process progresses (Arpit et al., 2017), they decrease the selected fraction of the minibatch over time.

Some theoretical justification for these approaches is provided by Zheng et al. (2020) in cases where the probability mass for instances very close to the decision boundary is low. For noisy labels with transition probabilities that are independent of the features, a sufficiently accurate model for predicting the true class-conditional probability $\eta(X) = \mathbb{P}\{Y = 1 \mid X\}$ can identify mislabeled samples. Based on that, they developed a likelihood-ratio test to decide whether a label in the training data should be flipped. Despite the theoretical foundation of their approach, they still need some empirical adjustments to make the method work in practice, e.g. they introduced an additional *retroactive loss* term in order to stabilize training.

**Post-Processing**  It is also possible to first train a scorer on the noisy data naively, from which a classifier adapted to a given rate of missing labels can be constructed by choosing an appropriate threshold. For a naive scorer that predicts the class probabilities for each data point, the corrected threshold is given in

Menon et al. (2015). Similarly, the inference procedure of probabilistic label trees can be adapted to take into account a propensity model (Wydmuch et al., 2021).

Other approaches are to try to design new losses specifically tailored to deal with missing labels, such as a group-lasso based formulation (Bucak et al., 2011).

**Positive-Unlabeled Learning**   Learning with missing labels is highly related to the problem of learning from positive and unlabeled (PU) data. This can be interpreted in two ways, the *censoring* setting which is identical to learning with missing labels, and the *case-control* setting in which the positive labels are drawn independently from the unlabeled data (Elkan and Noto, 2008). In the latter case the marginal of the true labels of the training and test data might be different. In that setting, instead of a noise rate, the class prior $\pi$ needs to be known (or estimated), then a corrected loss function can be determined as in Du Plessis et al. (2015). The appearing difficulties, that non-negativity and convexity need not be preserved in the new loss, are the same as in our setting (Kiryo et al., 2017; Chou et al., 2020).

**Semi-Supervised Learning**   A slightly different setting with missing labels is given by semi-supervised learning. Here, for each example the values of only a (known) subset of the labels are available, that is label can be one of three values 1 (positive), -1 (negative), and 0 (unknown). If the loss function decomposes over labels, then one strategy for coping with this situation, taken in Yu et al. (2014), is to only sum up the contributions where the label is known, i.e. the unknown labels are masked out.

## 10   Summary and Discussion

We have shown that the modelling of missing label learning problems using a mask variable provides an easy way of deriving unbiased estimators for both the binary and the multilabel setting, if labels go missing independently. These unbiased estimators are unique, and may show undesired properties: Even if the original loss function was convex and lower-bounded, the estimate can be non-convex and unbounded. Even in a pure evaluation setting, where these properties are not required, increasing variance as the propensity decreases poses a significant problem and may preclude the use of unbiased estimates.

As a mitigation, we propose to use convex upper-bounds. For the binary case we can write down a general solution given in (68). In the multilabel setting, we have considered four important cases that arise out of the multilabel reductions given in Menon et al. (2019). Particularly favourable among them is the Pick-all-Labels reduction, as it directly leads to a convex function. For its corresponding normalized form, we still can construct a convex upper-bound. In the One-vs-All case without normalization, the binary convex upper-bound can be applied, but finding a bound with normalization is still an open problem. An overview of the reductions is given in Table 5.

These results suggest that asymptotically, PaL reductions are preferable over OvA reductions. In practice, however, the situation is less clear. In our experiments we observed that unnormalized OvA produced the best results in terms of recall, even though this loss is in fact not consistent for recall. The most clear recommendation that can be drawn from our results is that if you want to use a normalized reduction, PaL-Norm is to be preferred over OvA-Norm because we currently lack a convex surrogate for the latter.

As suggested by theoretical results (Theorem 9) and corroborated by the experiments, missing labels lead to a shift in the bias-variance trade-off. If the data has missing labels, more regularization is required,

Table 5: Overview of multilabel loss reductions.

| Reduction | Base | Consistency | Convexity |
|---|---|---|---|
| One-vs-All | Binary | Precision | Upper-Bound |
| OvA-Norm | Binary | Recall | — |
| Pick-all-Labels | Multiclass | Precision | Yes |
| PaL-Norm | Multiclass | Recall | Upper-Bound |

irrespective of whether training uses vanilla-, unbiased-, or convex upper-bound losses. Looking more closely at the overfitting phenomenon, we found that the generalization error can be split into two parts: the difference between the empirical errors on the noisy and the clean (finite) data, and the difference between the clean empirical error and the true risk. We found that the overfitting to the specific noise pattern substantially exceeded the overfitting to the finite sample. Our findings agree with the observation of Arpit et al. (2017) who found that typical regularizers prevent a deep network from memorizing noisy examples, while not hindering the learning of patterns from clean instances.

Due to the uniqueness results, the problems mentioned above are unavoidable when using unbiased estimates. This suggests that further research should look into loss functions that allow tuning the trade-off between bias and variance. Having a slight bias in the loss function would usually not be problematic, especially considering that the propensity values which we have assumed to be given in this paper will in practice actually be only estimates, so that the unbiased estimates derived here will also have a slight bias due to misspecification.

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

## A    Per-Example Recall

Applying the general solution from Theorem 19 to the definition of per-example Recall as given in equation (66) results in

$$\text{Recall}(\mathbf{y}, \hat{\mathbf{y}}) = \left( \prod_{i \in \mathcal{I}(\mathbf{y})} \frac{p_i - 1}{p_i} \right) \cdot \sum_{\mathcal{J} \subset \mathcal{I}(\mathbf{y})} |\mathcal{J}|^{-1} \left( \sum_{k \in \mathcal{J}} \hat{y}_k \right) \prod_{j \in \mathcal{J}} (p_j - 1)^{-1}. \tag{75}$$

Note that the predictions $\hat{\mathbf{y}}$ usually are a sparse vector, e.g. when calculating recall@k, there are exactly $k$ nonzero entries. This may allow for a slightly more efficient calculation. We denote the set of correct predictions as $\mathcal{S} := \mathcal{I}(\mathbf{y}) \cap \mathcal{I}(\hat{\mathbf{y}})$ and the missed labels as $\mathcal{T} = \mathcal{I}(\mathbf{y}) \setminus \mathcal{S}$, such that $\mathcal{I}(\mathbf{y}) = \mathcal{S} \cup \mathcal{T}$ and the sum over subsets of $\mathcal{I}(\mathbf{y})$ can be written as a nested sum for $\mathcal{S}$ and $\mathcal{T}$. For convenience, we abbreviate the common factor as $c(\mathbf{y}) := \prod_{i \in \mathcal{I}(\mathbf{y})} \frac{p_i - 1}{p_i}$, and set $d(U) := \prod_{j \in \mathcal{U}} (p_j - 1)$. This results in

$$\text{Recall}(\mathbf{y}, \hat{\mathbf{y}}) = c(\mathbf{y}) \sum_{\substack{\mathcal{U} \subset \mathcal{T} \\ \mathcal{V} \subset \mathcal{S}}} \frac{|\mathcal{V}|}{|\mathcal{U}| + |\mathcal{V}|} \left( \prod_{j \in \mathcal{U}} \frac{1}{p_j - 1} \right) \left( \prod_{k \in \mathcal{V}} \frac{1}{p_k - 1} \right) = c(\mathbf{y}) \sum_{\mathcal{V} \subset \mathcal{S}} \frac{|\mathcal{V}|}{d(\mathcal{V})} \sum_{\mathcal{U} \subset \mathcal{T}} \frac{d(\mathcal{U})^{-1}}{|\mathcal{U}| + |\mathcal{V}|}. \tag{76}$$

The second sum is almost independent of the first: If the number of elements in $\mathcal{V}$ were constant, we could change the nested sums into a product of two single sums. Therefore, we collect terms based on the number of elements in $\mathcal{V}$ and rearrange to arrive at

$$\text{Recall}(\mathbf{y}, \hat{\mathbf{y}}) = c(\mathbf{y}) \sum_{s=1}^{|\mathcal{S}|} \sum_{\substack{\mathcal{V} \subset \mathcal{S} \\ |\mathcal{V}|=s}} \frac{s}{d(\mathcal{V})} \cdot \sum_{\mathcal{U} \subset \mathcal{T}} \frac{d(\mathcal{U})^{-1}}{|\mathcal{U}| + s} = c(\mathbf{y}) \sum_{s=1}^{|\mathcal{S}|} \left( \sum_{\mathcal{U} \subset \mathcal{T}} \frac{d(\mathcal{U})^{-1}}{|\mathcal{U}| + s} \right) \sum_{\substack{\mathcal{V} \subset \mathcal{S} \\ |\mathcal{V}|=s}} \frac{s}{d(\mathcal{V})}. \tag{77}$$

## B    Additional Theorems and Proofs

### B.1    Unbiased Estimates for Binary Setting

**Theorem 10** (Uniqueness). *Let $|\mathcal{Z}| \geq 2$, $p \in (0,1]$, $q \in (0,p]$, and $\mathcal{F} = \mathcal{M}(\{0,1\} \times \mathcal{Z}, \mathbb{R})$ be a set of functions. Let $\mathfrak{p} \colon \mathcal{F} \longrightarrow \mathcal{F}$ be an operator that maps a function to an unbiased estimate in the missing labels setting, such that for all $(Z, Y, Y^*) \in \mathcal{P}_p$ that fulfill the masking model with propensity $p$ and marginal $q = \mathbb{E}[Y]$, it holds*

$$\forall f^* \in \mathcal{F} : \mathbb{E}[f^*(Y^*, Z)] = \mathbb{E}[\mathfrak{p}(f^*)(Y, Z)]. \tag{28}$$

*Then, $\mathfrak{p}$ is related to the propensity scoring operator $\mathfrak{p}_p$ by*

$$\mathfrak{p}(f^*)(y, z) = \mathfrak{p}_p(f^*)(y, z) + (y - q)\gamma \tag{29}$$

*for some $\gamma \in \mathbb{R}$.*

*Proof.* The fact that $\mathfrak{P}_p$ fulfills the condition (28) follows from Theorem 7. Because $\mathbb{E}[Y] = q$, the additional term has expectation zero.

Let $\mathfrak{P}$ be another operator for which the condition holds, then (28) is in particular also fulfilled for distributions of the form

$$X, Y^* \sim q^* \delta(x_1, 1) + (1 - q^*)\delta(x_2, 0), \tag{78}$$

where $\delta(x, y)$ denotes the Dirac measure of point $(x, y)$ and $q^* := q/p$. These are valid because $q \overset{!}{=} \mathbb{P}\{Y = 1\} = \mathbb{P}\{Y^* = 1, M = 1\} = p \cdot q^* = q$.

Let us further write $f = \mathfrak{P}(f^*)$, $f^*(y, x) = yg^*(x) + h^*(x)$ and $f(x, y) = yg(x) + h(x)$. Since $y \in \{0, 1\}$, the decomposition into $g$ and $h$ is always possible.

In this notation, we can explicitly calculate the expectations

$$\mathbb{E}[f^*(Y^*, X)] = \mathbb{E}[Y^* g^*(X) + h^*(X)] = q^*(g^*(x_1) + h^*(x_1)) + (1 - q^*)h^*(x_2) \tag{79}$$

$$\mathbb{E}[f(Y, X)] = \mathbb{E}[MY^* g(X) + h(X)] = q^*(pg(x_1) + h(x_1)) + (1 - q^*)h(x_2). \tag{80}$$

By assumption on $\mathfrak{P}$ these two are equal:

$$q^*(g^*(x_1) + h^*(x_1)) + (1 - q^*)h^*(x_2) = q^*(pg(x_1) + h(x_1)) + (1 - q^*)h(x_2) \tag{81}$$

$$q^*(g^*(x_1) + h^*(x_1) - h(x_1)) + (1 - q^*)(h^*(x_2) - h(x_2)) = p\, q^* g(x_1). \tag{82}$$

Setting $x_1 = x_2 =: x$ gives

$$q^*(g^*(x) - pg(x)) = h(x) - h^*(x), \tag{83}$$

which can be plugged back into (82)

$$q^*(g^*(x_1) - q^*(g^*(x_1) - pg(x_1))) + (1 - q^*)(q^*(g^*(x_2) - pg(x_2))) = pq^* g(x_1) \tag{84}$$

$$(1 - q^*)g^*(x_1) + (q^* - 1)pg(x_1) + (1 - q^*)(g^*(x_2) - pg(x_2)) = 0 \tag{85}$$

$$g^*(x_1) - pg(x_1) + g^*(x_2) - pg(x_2) = 0. \tag{86}$$

Since this equation holds for all $x_1, x_2 \in \mathcal{X}$, it determines $g$ up to a constant shift. Let $x_0 \in \mathcal{X}$ be fixed and denote $c = g^*(x_0)/p - g(x_0)$, then for arbitrary $x \in \mathcal{X}$

$$g(x) = g^*(x)/p + c. \tag{87}$$

Putting this back into (83) gives

$$h(x) = h^*(x) - pcq^* = h^*(x) - qc. \tag{88}$$

Combining these two expressions shows the claim

$$\mathfrak{P}(f^*)(y, x) = yg(x) + h(x) = \frac{yg^*(x)}{p} + yc + h^*(x) - qc = \frac{y}{p}g^*(x) + h^*(x) + (y - q)c. \tag{89}$$

$\square$

## B.2 Generalization Bound

In this section we present a proof for Theorem 9. To that end, we first proof some helper results as presented below:

**Theorem 27.** *Let $\mathcal{H} \subset \mathcal{M}(\mathcal{X}, \hat{\mathcal{Y}})$ be a function class, and $f \colon \mathcal{Y} \times \hat{\mathcal{Y}} \longrightarrow [a, b]$ be a bounded, $\rho$-Lipschitz continuous (in the second argument) function. Denote $\tilde{f} := \mathfrak{P}_p f$. Given a sample of $n$ noisy training points, it holds with probability of at least $1 - \delta$ that*

$$\sup_{h \in \mathcal{H}} \left( \hat{\mathrm{R}}_{\tilde{f}}[h] - \mathrm{R}_{\tilde{f}}[h] \right) \leq \frac{4 - 2p}{p} \rho \, \mathfrak{R}_n(\mathcal{H}) + \frac{(2 - p)(b - a)}{p} \sqrt{\frac{\log(1/\delta)}{2n}} \tag{90}$$

$$\sup_{h \in \mathcal{H}} \left( \mathrm{R}_{\tilde{f}}[h] - \hat{\mathrm{R}}_{\tilde{f}}[h] \right) \leq \frac{4 - 2p}{p} \rho \, \mathfrak{R}_n(\mathcal{H}) + \frac{(2 - p)(b - a)}{p} \sqrt{\frac{\log(1/\delta)}{2n}}. \tag{91}$$

*Proof.* First, we determine the Lipschitz-constant of $\tilde{f}$. For $y = 0$, it is the same as that of $f$, so we only need to consider the $y = 1$ case.

$$\left| \tilde{f}(1, x_1) - \tilde{f}(1, x_2) \right| = \frac{f(1, x_1) + (p - 1)f(0, x_1) - f(1, x_2) - (p - 1)f(0, x_2)}{p} \tag{92}$$

$$\leq \frac{1}{p} |f(1, x_1) - f(1, x_2)| + \frac{1 - p}{p} |f(0, x_1) - f(0, x_2)| \tag{93}$$

$$\leq \left( \frac{1}{p} + \frac{1 - p}{p} \right) \rho \|x_1 - x_2\|. \tag{94}$$

In (93) we made use of the fact that $0 < p \leq 1$. This also implies that $\frac{2-p}{p} \geq 1$, and thus the Lipschitz constant of $\tilde{f}$ is given by $\frac{2-p}{p}\rho$.

Next we calculate the range of $\tilde{f}$. We have $\forall x \in \mathcal{X}$, that

$$\tilde{a} := \frac{a + (p-1)b}{p} \leq a \leq \tilde{f}(1, x) \leq b \leq \frac{b + (p-1)a}{p} =: \tilde{b}. \tag{95}$$

Here, the first and last inequality follow from $p \in (0, 1]$ and $a \leq b$. Define $c := \tilde{b} - \tilde{a}$.

Finally, we can construct a function $f_{01} : \mathcal{Y} \times \hat{\mathcal{Y}} \longrightarrow [0, 1]$ by the affine transformation $f_{01} = c^{-1}(\tilde{f} - \tilde{a})$ such that

$$\mathrm{R}_{\tilde{f}}[h] - \hat{\mathrm{R}}_{\tilde{f}}[h] = c\left(\mathrm{R}_{f_{01}}[h] - \hat{\mathrm{R}}_{f_{01}}[h]\right). \tag{96}$$

The right hand side can be bounded with probability $1 - \delta$ using Mohri et al. (2018, Theorem 3.3) by

$$\mathrm{R}_{f_{01}}[h] - \hat{\mathrm{R}}_{f_{01}}[h] \leq \mathfrak{R}_n(f_{01} \circ \mathcal{H}) + \sqrt{\frac{\log(1/\delta)}{2n}}. \tag{97}$$

As the Lipschitz-constant of $f_{01}$ is $c^{-1}p^{-1}(2-p)$, by the contraction lemma (Shalev-Shwartz and Ben-David, 2014, Lemma 26.9) we have

$$\mathfrak{R}_n(f_{01} \circ \mathcal{H}) \leq c^{-1}\frac{2-p}{p}\rho \,\mathfrak{R}_n(\mathcal{H}). \tag{98}$$

Thus with probability $1 - \delta$ and $\forall h \in \mathcal{H}$

$$\mathrm{R}_{\tilde{f}}[h] - \hat{\mathrm{R}}_{\tilde{f}}[h] \leq cc^{-1}\frac{2-p}{p}\rho\,\mathfrak{R}_n(\mathcal{H}) + c\sqrt{\frac{\log(1/\delta)}{2n}} \tag{99}$$

$$= \frac{2-p}{p}\rho\,\mathfrak{R}_n(\mathcal{H}) + \frac{(2-p)(b-a)}{p}\sqrt{\frac{\log(1/\delta)}{2n}} \tag{100}$$

The second bound follows by replacing $f$ with $-f$. $\qquad\square$

This result is very similar to Natarajan et al. (2017, Lemma 8). However, that theorem is missing a scaling factor with the range of the loss function, as argued below. For reference, the original statement of the theorem is

**Theorem 28** (Natarajan et al. (2017, Lemma 8)). *Let $l(t, y)$ be $L$-Lipschitz in $t$ (for every $y$). Then, for any $\alpha \in (0, 1)$, with probability at least $1 - \delta$,*

$$\max_{f \in \mathcal{F}} \left|\hat{R}_{\tilde{l}_\alpha}(f) - R_{\tilde{l}_\alpha, D_\rho}(f)\right| \leq 2L_\rho \mathfrak{R}_n(\mathcal{F}) + \sqrt{\frac{\log(1/\delta)}{2n}}, \tag{101}$$

*where $L_\rho \leq 2L/(1 - \rho_{+1} - \rho_{-1})$ is the Lipschitz constant of $\tilde{l}_\alpha$.*

In the first step of the proof, they invoke a "Basic Rademacher bound between risks and empirical risks" that states

$$\max_{f \in \mathcal{F}} \left|\hat{R}_{\tilde{l}_\alpha}(f) - R_{\tilde{l}_\alpha, D_\rho}(f)\right| \leq 2\,\mathfrak{R}_n(\tilde{l}_\alpha \circ \mathcal{F}) + \sqrt{\frac{\log(1/\delta)}{2n}} \tag{102}$$

However, such a bound either requires the range of $\tilde{l}_\alpha$ to be a subset of $[0, 1]$ (Mohri et al., 2018, Thm 3.3), or introduces an additional factor in front of the square root term as in Shalev-Shwartz and Ben-David (2014, Thm 26.5). Also, they are using a two-sided bound instead of a one-sided one as in the two cited theorems, which means that $\delta$ needs to be replaced with $\delta/2$ because the square-root term comes from an application of Mc. Diamids inequality.

**Lemma 29.** *For any $f \in \mathcal{M}(\mathcal{Y} \times \mathcal{X}, \mathbb{R})$, $p \in (0,1]$ and $\mathbb{E}[Y] = q$, it holds*

$$\mathbb{E}[|\mathfrak{P}_p(f)(Y, X) - f(Y, X)|] \leq q \frac{1-p}{p} m \quad with \quad m := \sup_{x \in \mathcal{X}} (|f(1, x) - f(0, x)|). \tag{103}$$

*Proof.* For notational convenience denote $U := f(1, X)$ and $V := f(0, X)$. Substituting $\mathfrak{P}_p(f)$, difference can be simplified to

$$\begin{aligned}
\mathfrak{P}_p(f)&(Y, X) - f(Y, X) \\
&= Yp^{-1}\left(f(1, X) + (p-1)f(0, X)\right) + (1-Y)f(0, X) - f(Y, X) \\
&= Yp^{-1}\left(U + (p-1)V\right) + (1-Y)V - YU - (1-Y)V \\
&= (p^{-1} - 1)(YU) + \frac{p-1}{p}YV = \frac{1-p}{p}(Y(U-V)).
\end{aligned}$$

As by definition of $m$ it holds that $|U - V| \leq m$, the expectation is bounded by

$$\mathbb{E}[|\mathfrak{P}_p(f)(Y, X) - f(Y, X)|] = \frac{1-p}{p} \mathbb{E}[|Y(U-V)|] \leq \frac{1-p}{p} \mathbb{E}[Ym] \leq \frac{1-p}{p} qm. \tag{104}$$

$\square$

**Theorem 9** (Generalization bounds). *Let $\mathcal{H}$ be a function class with Rademacher complexity $\mathfrak{R}_n(\mathcal{H})$. Let $f^*: \mathcal{Y} \times \hat{\mathcal{Y}} \longrightarrow [a, b]$ for $a < b \in \mathbb{R}$ be a function that is $\rho$-Lipschitz continuous in its second argument. Let $f := \mathfrak{p}_p(f^*)$ and denote*

$$r^\star := \inf_{h \in \mathcal{H}} \mathrm{R}^*_{f^*}[h], \quad \hat{h} := \operatorname*{argmin}_{h \in \mathcal{H}} \hat{\mathrm{R}}_f[h], \quad \tilde{h} := \operatorname*{argmin}_{h \in \mathcal{H}} \hat{\mathrm{R}}_{f^*}[h]. \tag{23}$$

*as well as*

$$\begin{aligned}
c &:= \rho \, \mathfrak{R}_n(\mathcal{H}) + (b-a)\sqrt{\frac{\log(2/\delta)}{2n}} \\
m &:= \sup_{z \in \mathcal{Z}} (|f^*(1, z) - f^*(0, z)|).
\end{aligned} \tag{24}$$

*For a given sample of $n$ points, it holds with probability at least $1 - \delta$*

$$\hat{r} := \mathrm{R}^*_{f^*}\left[\hat{h}\right] \qquad \leq r^\star \qquad\qquad\qquad + 2\frac{2-p}{p}c \tag{25}$$

$$\tilde{r} := \mathrm{R}^*_{f^*}\left[\tilde{h}\right] \qquad \leq r^\star + \qquad\qquad q\frac{1-p}{p}m \qquad + 2c, \tag{26}$$

*where $q := \mathbb{E}[Y] \leq 1$.*

*Proof.* Let $\epsilon > 0$ and choose $h'$ such that

$$wever \ \mathrm{R}^*_{f^*}[h'] \leq r' + \epsilon. \tag{105}$$

From this follows $\hat{r} - r' \leq \mathrm{R}^*_{f^*}\left[\hat{h}\right] - \mathrm{R}^*_{f^*}[h'] + \epsilon$ and $\tilde{r} - r' \leq \mathrm{R}^*_{f^*}\left[\tilde{h}\right] - \mathrm{R}^*_{f^*}[h'] + \epsilon$.

We can apply Theorem 27 to the function class $\{h'\}$ using that $\mathfrak{R}_n(\{h' = 0\}) = 0$ to get with probability $1 - \delta/2$

$$\hat{\mathrm{R}}_f[h'] - \mathrm{R}_f[h'] \leq \frac{(2-p)(b-a)}{p}\sqrt{\frac{\log(2/\delta)}{2n}}. \tag{106}$$

For the first inequality, we can use the unbiasedness of $f$, and the near optimality of $\hat{h}$ regarding $\hat{R}_f$ to bound

$$
\begin{aligned}
R_{f^*}^*\left[\hat{h}\right] - R_{f^*}^*[h'] &= R_f\left[\hat{h}\right] - R_f[h'] && \text{(unbiasedness)} \\
&= R_f\left[\hat{h}\right] - \hat{R}_f\left[\hat{h}\right] + \hat{R}_f\left[\hat{h}\right] - \hat{R}_f[h'] + \hat{R}_f[h'] - R_f[h'] \\
&\leq R_f\left[\hat{h}\right] - \hat{R}_f\left[\hat{h}\right] + \hat{R}_f[h'] - R_f[h'] && \text{(optimality)} \\
&\leq \sup_{h\in\mathcal{H}}\left(R_f[h] - \hat{R}_f[h]\right) + \hat{R}_f[h'] - R_f[h'].
\end{aligned}
$$

Applying a union bound to the remaining two terms, with probability $1-\delta$

$$
\begin{aligned}
\hat{r} - r' &\leq \epsilon + \frac{4-2p}{p}\,\mathfrak{R}(\mathcal{H}) + \frac{(2-p)(b-a)}{p}\sqrt{\frac{\log(2/\delta)}{2n}} + \frac{(2-p)(b-a)}{p}\sqrt{\frac{\log(2/\delta)}{2n}} \\
&= \epsilon + \frac{4-2p}{p}\left(\mathfrak{R}_n(\mathcal{H}) + (b-a)\sqrt{\frac{\log(2/\delta)}{2n}}\right).
\end{aligned}
\tag{107}
$$

With $\epsilon \to 0$ the first claim follows.

For $\tilde{h}$ we can decompose the risk difference into

$$
R_{f^*}^*[\tilde{h}] - R_{f^*}^*[h'] = \underbrace{R_{f^*}^*[\tilde{h}] - \hat{R}_{f^*}[\tilde{h}]}_{a} + \underbrace{\hat{R}_{f^*}[\tilde{h}] - \hat{R}_{f^*}[h']}_{b} + \underbrace{\hat{R}_{f^*}[h'] - R_{f^*}^*[h']}_{c}
\tag{108}
$$

and look at the contributions separately. Because $\tilde{h}$ is an ERM, we have

$$
b = \hat{R}_{f^*}[\tilde{h}] - \hat{R}_{f^*}[h'] \leq 0.
\tag{109}
$$

Further, using the unbiasedness and applying Lemma 29 to the function $(x,y) \mapsto f(y, \tilde{h}(x))$

$$
\begin{aligned}
a = R_{f^*}^*[\tilde{h}] - \hat{R}_{f^*}[\tilde{h}] &= R_f[\tilde{h}] - \hat{R}_{f^*}[\tilde{h}] && \text{(unbiasedness)} \\
&= R_f[\tilde{h}] - R_{f^*}[\tilde{h}] + R_{f^*}[\tilde{h}] - \hat{R}_{f^*}[\tilde{h}] && (110) \\
&\leq q\frac{1-p}{p}m + R_{f^*}[\tilde{h}] - \hat{R}_{f^*}[\tilde{h}] && \text{(Lemma 29)} \\
&\leq q\frac{1-p}{p}m + \sup_{h\in\mathcal{H}}\left(R_{f^*}[h] - \hat{R}_{f^*}[h]\right) && (111)
\end{aligned}
$$

We can apply Theorem 27 with $p = 1$ to get corresponding bounds for $f^* = \mathfrak{P}_0 f^*$ so that with probability $1 - \delta/2$ each

$$
\sup_{h\in\mathcal{H}}\left(R_{f^*}[h] - \hat{R}_{f^*}[h]\right) \leq \frac{4-2}{1}\,\mathfrak{R}_n(\mathcal{H}) + \frac{(2-1)(b-a)}{1}\sqrt{\frac{\log(2/\delta)}{2n}}
\tag{112}
$$

$$
c = \hat{R}_{f^*}[h'] - R_{f^*}^*[h'] \leq \frac{(2-1)(b-a)}{1}\sqrt{\frac{\log(2/\delta)}{2n}}.
\tag{113}
$$

Thus, by union bound, with probability $1 - \delta$, it holds that

$$
R_{f^*}^*[\tilde{h}] - R_{f^*}^*[h'] \leq q\frac{1-p}{p}m + 2\,\mathfrak{R}_n(\mathcal{H}) + 2(b-a)\sqrt{\frac{\log(2/\delta)}{2n}} + \epsilon.
\tag{114}
$$

Letting $\epsilon \to 0$ proves the claim. $\qquad\square$

### B.3 Unbiased Estimates for Multilabel Setting

**Theorem 19** (Multilabel Loss)**.** *Let $(Z, \boldsymbol{Y}^*, \boldsymbol{Y}) \in \mathcal{P}_{\boldsymbol{p}}^{\perp\!\!\!\perp}(\mathcal{Z})$, and $f^*\colon \mathcal{Y}^m \times \mathcal{Z} \longrightarrow \mathbb{R}$. An unbiased version of $f^*$ can be calculated using the propensity-scored expression $\mathfrak{p}_{\boldsymbol{p}}(f^*) = f$ given by*

$$f\colon (\boldsymbol{y}, z) \mapsto \sum_{\boldsymbol{y}' \preceq \boldsymbol{y}} \left( \prod_{j:y_j=1} \frac{y_j'(2-p_j) + p_j - 1}{p_j} \right) f^*(\boldsymbol{y}', z) \tag{53}$$

*where $\boldsymbol{y}' \preceq \boldsymbol{y}$ means $\{0, 1\} \ni y_j' \leq y_j$.*

*Proof.* As in the binary case, we can use the finiteness of $\{0,1\}^l$ to write $f^*$ as

$$f^*(y, x) = \sum_{\mathcal{I} \subset [l]} \mathbb{1}\left\{ y = \boldsymbol{1}^{(\mathcal{I})} \right\} f^*(\boldsymbol{1}^{(\mathcal{I})}, x) \tag{115}$$

$$= \sum_{\mathcal{I} \subset [l]} f^*(\boldsymbol{1}^{(\mathcal{I})}, x) \left( \prod_{i \in \mathcal{I}} y_i \right) \prod_{j \in \overline{\mathcal{I}}} (1 - y_j) \tag{116}$$

by rewriting the indicator function as products of $y_i$ and $1 - y_j$.

First, we show the unbiasedness for the expression

$$\tilde{f}(\mathbf{y}, x) \coloneqq \sum_{\mathcal{I} \subset [l]} f^*(\boldsymbol{1}^{(\mathcal{I})}, x) \left( \prod_{i \in \mathcal{I}} \frac{y_i}{p_i} \right) \prod_{j \in \overline{\mathcal{I}}} \left( 1 - \frac{y_j}{p_j} \right). \tag{117}$$

Later, we will show that this is in fact equivalent to (53). Using the linearity of the expectation we can explicitly calculate

$$\mathbb{E}\big[\tilde{f}(\mathbf{Y}, X)\big] = \sum_{\mathcal{I} \subset [l]} \mathbb{E}\left[ f^*(\boldsymbol{1}^{(\mathcal{I})}, X) \left( \prod_{i \in \mathcal{I}} \frac{M_i Y_i^*}{p_i} \right) \prod_{j \in \overline{\mathcal{I}}} \left( 1 - \frac{M_j Y_j^*}{p_j} \right) \right] \tag{118}$$

For a fixed subset $\mathcal{I}$ we can rewrite

$$f^*(\boldsymbol{1}^{(\mathcal{I})}, X) \left( \prod_{i \in \mathcal{I}} \frac{M_i Y_i^*}{p_i} \right) \prod_{j \in \overline{\mathcal{I}}} \left( 1 - \frac{M_j Y_j^*}{p_j} \right) = \sum_{\mathcal{J} \subset [l]} f^*(\boldsymbol{1}^{(\mathcal{I})}, X) \prod_{i \in \mathcal{J}} \alpha_{\mathcal{J}} M_i Y_i^* \tag{119}$$

for appropriately chosen coefficients $\alpha_{\mathcal{J}} \in \mathbb{R}$. Using the independence of $\mathbf{M}$, it follows that

$$\mathbb{E}\left[ f^*(\boldsymbol{1}^{(\mathcal{I})}, X) \prod_{i \in \mathcal{J}} \alpha_{\mathcal{J}} M_i Y_i^* \right] = \mathbb{E}\left[ \prod_{i \in \mathcal{J}} M_i \right] \mathbb{E}\left[ f^*(\boldsymbol{1}^{(\mathcal{I})}, X) \prod_{j \in \mathcal{J}} \alpha_{\mathcal{J}} Y_j^* \right] \tag{120}$$

$$= \left( \prod_{i \in \mathcal{J}} q_i \right) \mathbb{E}\left[ f^*(\boldsymbol{1}^{(\mathcal{I})}, X) \prod_{j \in \mathcal{J}} \alpha_{\mathcal{J}} Y_j^* \right] = \mathbb{E}\left[ f^*(\boldsymbol{1}^{(\mathcal{I})}, X) \prod_{j \in \mathcal{J}} \alpha_{\mathcal{J}} p_j Y_j^* \right]. \tag{121}$$

Therefore, we can replace all occurrences of $M_i$ in the expectation with $p_i$ and compare the result to (116)

$$\mathbb{E}\big[\tilde{f}(\mathbf{Y}, X)\big] = \sum_{\mathcal{I} \subset [l]} \mathbb{E}\left[ f^*(\boldsymbol{1}^{(\mathcal{I})}, X) \left( \prod_{i \in \mathcal{I}} \frac{p_i Y_i^*}{p_i} \right) \prod_{j \in \overline{\mathcal{I}}} \left( 1 - \frac{p_j Y_j^*}{p_j} \right) \right] \tag{122}$$

$$= \sum_{\mathcal{I} \subset [l]} \mathbb{E}\left[ f^*(\boldsymbol{1}^{(\mathcal{I})}, X) \left( \prod_{i \in \mathcal{I}} Y_i^* \right) \prod_{j \in \overline{\mathcal{I}}} (1 - Y_j^*) \right] \tag{123}$$

$$= \mathbb{E}[f^*(\mathbf{Y}^*, X)]. \tag{124}$$

Now we show that $\tilde{f} = f$: For any $\mathcal{J} \not\subset \mathcal{I}(\mathbf{y})$ there is a $j \in \mathcal{J}$ such that $y_j = 0$, so the contribution of that summand is zero. Therefore

$$\tilde{f}(\mathbf{y}, x) = \sum_{\mathcal{J} \subset \mathcal{I}(\mathbf{y})} f^*(\mathbf{1}^{(\mathcal{J})}, x) \left( \prod_{i \in \mathcal{J}} \frac{1}{p_i} \right) \prod_{j \in \overline{\mathcal{J}}} \left( 1 - \frac{y_j}{p_j} \right). \tag{125}$$

Now, for every $j \in \overline{\mathcal{I}(\mathbf{y})}$ we know that $y_j = 0$, so we can simplify further

$$= \sum_{\mathcal{J} \subset \mathcal{I}(\mathbf{y})} f^*(\mathbf{1}^{(\mathcal{J})}, x) \left( \prod_{i \in \mathcal{J}} \frac{1}{p_i} \right) \prod_{k \in \mathcal{I}(\mathbf{y}) \setminus \mathcal{J}} \left( 1 - \frac{1}{p_k} \right). \tag{126}$$

Finally, note that

$$\left( \prod_{i \in \mathcal{J}} \frac{1}{p_i} \right) \prod_{k \in \mathcal{I}(\mathbf{y}) \setminus \mathcal{J}} \left( 1 - \frac{1}{p_k} \right) = \left( \prod_{i \in \mathcal{J}} \frac{1}{p_i} \left( \frac{p_i - 1}{p_i} \right)^{-1} \right) \prod_{k \in \mathcal{I}(\mathbf{y})} \left( \frac{p_k - 1}{p_k} \right), \tag{127}$$

which proves the statement. $\qquad\square$

**Theorem 20** (Multilabel Uniqueness). *Then debiasing operator $\mathfrak{p}_{\boldsymbol{p}}$ from Theorem 19 is unique, i.e., for any $f^* \colon \mathcal{Y}^m \times \mathcal{Z} \longrightarrow \mathbb{R}$, any unbiased version $f$ is equal to $\mathfrak{p}_{\boldsymbol{p}}(f^*)$. Formally, if for all $(Z, \boldsymbol{Y}^*, \boldsymbol{Y}) \in \mathcal{P}_{\boldsymbol{p}}^{\perp\!\!\!\perp}(\mathcal{Z})$ it holds that*

$$\mathbb{E}[f^*(\boldsymbol{Y}^*, Z)] = \mathbb{E}[f(\boldsymbol{Y}, Z)], \tag{56}$$

*then $f = \mathfrak{p}_{\boldsymbol{p}}(f^*)$.*

*Proof.* Let $f^*$ be an arbitrary function $f^* \colon \mathcal{Y} \times \mathcal{X} \longrightarrow \mathbb{R}$. We need to show that

$$\tilde{f} := \mathfrak{P}(f^*) = \mathfrak{P}_{\mathbf{p}}(f^*) =: f. \tag{128}$$

Since (56) needs to work for all possible distributions of $X$ and $\boldsymbol{Y}^*$, it needs to work in particular also for $\mathbb{P}\{X = x, \boldsymbol{Y}^* = \mathbf{y}^*\} = 1$. Since $\boldsymbol{Y}$ can take only finitely many states, we can decompose $\mathfrak{P}(f^*)$ into a sum over these states. Since $f$ is known to fulfill (56), the equation for $\tilde{f}$ becomes

$$0 = \mathbb{E}\big[\tilde{f}(\mathbf{Y}, X) - f(\mathbf{Y}, X)\big] = \sum_{\mathbf{y} \in 2^{\mathcal{Y}}} \mathbb{E}\big[(\tilde{f}(x, \mathbf{y}) - f(x, \mathbf{y}))\mathbb{1}\{\mathbf{y} = Y\}\big] = \sum_{\mathbf{y}' \in 2^{\mathcal{Y}}} (\tilde{f}(x, \mathbf{y}') - f(x, \mathbf{y}'))\mathbb{P}\{\mathbf{y}' = Y\} \tag{129}$$

Next we show that for all $\mathbf{y} \in \mathcal{Y}$ it holds that $\tilde{f}(x, \mathbf{y}) - f(x, \mathbf{y}) = 0$ via induction. First, assume that $\mathbf{y}^* = \mathbf{0}$. Then $\mathbb{P}\{\mathbf{y}' = Y\} = \mathbb{1}\{\mathbf{y}' = \mathbf{0}\}$, and (129) simplifies to

$$0 = \tilde{f}(x, \mathbf{0}) - f(x, \mathbf{0}), \tag{130}$$

and therefore $\tilde{f}$ and $f$ are equal for this $\mathbf{y}^*$.

Next, assume that $\tilde{f}(\mathbf{y}', x) = f(\mathbf{y}', x)$ for all $\mathbf{y}'$ that have at most $m$ nonzero entries. Let $\mathbf{y}$ be a vector with $m + 1$ nonzero entries, then (129) can be written as

$$0 = \sum_{\substack{\mathbf{y}' \in 2^{\mathcal{Y}} \\ |\mathbf{y}'| \leq m}} (\tilde{f}(x, \mathbf{y}') - f(x, \mathbf{y}'))\mathbb{P}\{\mathbf{y}' = Y\} + (\tilde{f}(x, \mathbf{y}^*) - f(x, \mathbf{y}^*))\mathbb{P}\{\mathbf{y}^* = Y\}$$
$$= (\tilde{f}(x, \mathbf{y}^*) - f(x, \mathbf{y}^*))\mathbb{P}\{\mathbf{y}^* = Y\}. \tag{131}$$

Here, the first sum vanishes because all the summands are using a vector $\mathbf{y}'$ with at most $m$ elements. Because we assume that all propensities are nonzero, the $\mathbb{P}\{\mathbf{y}^* = Y\}$ factor is nonzero, which implies that $\tilde{f}(x, \mathbf{y}^*) - f(x, \mathbf{y}^*)$ has to be zero.

Therefore, $\tilde{f}$ and $f$ are identical. Since this holds for all $x \in \mathcal{X}$, the operators $\mathfrak{P}$ and $\mathfrak{P}_{\mathbf{p}}$ have to be identical. $\qquad\square$

## B.4 Multilabel upper bound

**Theorem 25** (Normalized Label Upper-Bound). *Let $(Z, \boldsymbol{Y}^*, \boldsymbol{Y}) \in \mathcal{P}_{\boldsymbol{p}}^{\perp\!\!\!\perp}(\mathcal{Z})$, replacing the true labels with the unbiased estimate of the observed labels as shown in Equation 69 results in an upper bound, whose error itself can be bounded by a data-dependent term*

$$\mathbb{E}[T_i^*] + \sum_{j \neq i} \left( \frac{1 - p_j}{p_j} \right) \mathbb{E}\left[ \frac{Y_i}{p_i} \cdot \frac{Y_j}{p_j} \right] \geq \mathbb{E}[\tilde{T}_i] \geq \mathbb{E}[T_i^*]. \tag{70}$$

*Proof.* For convenience denote $S_i^* \coloneqq \sum_{j \neq i} Y_j^*$ and $S_i \coloneqq \sum_{j \neq i} Y_j / p_j$, and note that $S_i$, being a function of $\boldsymbol{Y}_{\neg i}^*$ and $\boldsymbol{M}_{\neg i}$, is independent of $M_i$. By pulling out known factors and using the independence of $\boldsymbol{M}$ and $\boldsymbol{Y}^*$ we can show that

$$\mathbb{E}[S_i \mid \boldsymbol{Y}^*] = \sum_{j \neq i} \mathbb{E}[M_j Y_j^* / p_j \mid \boldsymbol{Y}^*] = \sum_{j \neq i} Y_j^* \, \mathbb{E}[M_j / p_j \mid \boldsymbol{Y}^*] = S_i^*. \tag{132}$$

Expanding terms and using independence of $M_i$, then applying the tower property and pulling out the measurable factor results in

$$\mathbb{E}[\tilde{T}_i] = \mathbb{E}\left[ \frac{M_i Y_i^* / p_i}{1 + S_i} \right] = \mathbb{E}\left[ \frac{M_i}{p_i} \right] \mathbb{E}\left[ \frac{Y_i^*}{1 + S_i} \right] = \mathbb{E}\left[ \mathbb{E}\left[ \frac{Y_i^*}{1 + S_i} \,\bigg|\, \boldsymbol{Y}^* \right] \right] = \mathbb{E}\left[ Y_i^* \, \mathbb{E}\left[ \frac{1}{1 + S_i} \,\bigg|\, \boldsymbol{Y}^* \right] \right].$$

The function $h \colon _{\geq 0} \longrightarrow$ given by $t \mapsto 1 / (1 + t)$ is convex, because its second derivative is $2(1 + t)^{-3}$, which is larger than zero for non-negative $t$. Because $S_i \geq 0$ almost surely, we can apply Jensen's inequality to the inner expectation and use (132)

$$\mathbb{E}[\tilde{T}_i] \geq \mathbb{E}\left[ \frac{Y_i^*}{1 + \mathbb{E}[S_i \mid \boldsymbol{Y}^*]} \right] = \mathbb{E}\left[ \frac{Y_i^*}{1 + S_i^*} \right] = \mathbb{E}[T_i^*].$$

On the other hand we can use the Taylor formula with intermediate point $\zeta \in [S_i, S_i^*]$ to expand

$$\frac{1}{1 + S_i} = \frac{1}{1 + S_i^*} - \frac{S_i - S_i^*}{(1 + S_i^*)^2} + \frac{(S_i - S_i^*)^2}{(1 + \zeta)^3}. \tag{133}$$

Using $\zeta \geq 0$ to bound the denominator of the last term, then multiplying with $Y_i^*$ and taking the expectation gives

$$\mathbb{E}\left[ \frac{Y_i^*}{1 + S_i} \right] \leq \mathbb{E}\left[ \frac{Y_i^*}{1 + S_i^*} \right] + \mathbb{E}[Y_i^* (S_i - S_i^*)^2]. \tag{134}$$

The variance term can be calculated by substituting $S_i$ and $S_i^*$, expanding the sum, and using the independence of $M$ to show that the mixed terms are zero:

$$\begin{aligned}
\mathbb{E}[Y_i^* (S_i - S_i^*)^2] &= \mathbb{E}\left[ Y_i^* \left( \sum_{j \neq i} Y_j^* \left( \frac{M_j}{p_j} - 1 \right) \right)^2 \right] \\
&= \sum_{j \neq i} \mathbb{E}\left[ Y_i^* (Y_j^*)^2 \left( \frac{M_j}{p_j} - 1 \right)^2 \right] + \sum_{j \neq i} \sum_{k \notin \{i, j\}} \mathbb{E}[Y_i^* Y_j^* Y_k^*] \, \mathbb{E}\left[ \frac{M_j}{p_j} - 1 \right] \mathbb{E}\left[ \frac{M_k}{p_k} - 1 \right] \\
&= \sum_{j \neq i} \mathbb{E}[Y_i^* Y_j^*] \, \mathbb{E}\left[ \frac{M_j}{p_j^2} - 2 \frac{M_j}{p_j} + 1 \right] = \sum_{j \neq i} \left( \frac{1 - p_j}{p_j} \right) \mathbb{E}\left[ \frac{Y_i}{p_i} \cdot \frac{Y_j}{p_j} \right]. \quad (135)
\end{aligned}$$

$\square$

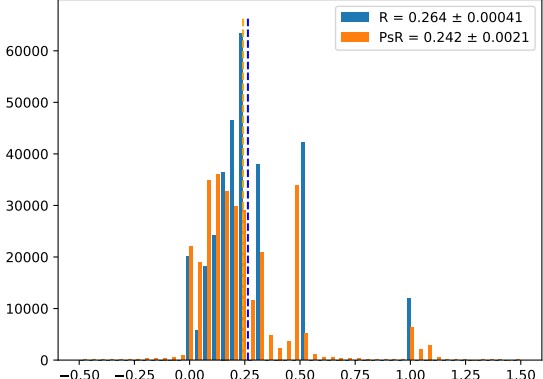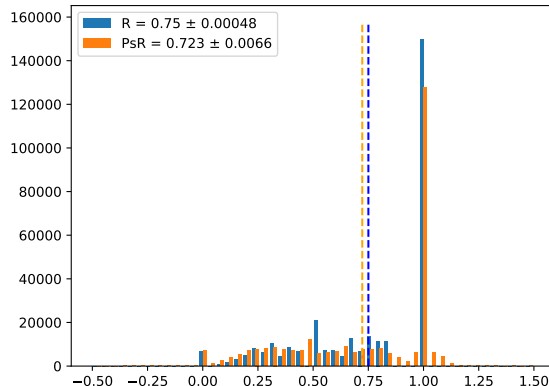

Figure 7: Histograms of propensity-scored (PsR) and vanilla recall (R) for top-1 (left) and top-5 (right) predictions for the `AmazonCat-13K` dataset for a DiSMEC model trained with a convex surrogate of the propensity-scored 0-1 loss. The y-axis denotes the number of examples in the test set for which the estimate falls into the corresponding bin. The errors have been calculated by boostrapping a 95 % confidence interval. The dashed vertical lines denote the mean.

## C    Experiments

### C.1    PsRecall estimation

The networks used to generate the results in Table 3 were trained using the DiSMEC (Babbar and Schölkopf, 2017) algorithm, with the loss function being either the squared-hinge-loss (VN) or a squared-hinge-loss based convex surrogate of the unbiased estimate of the 0-1 loss as described in Qaraei et al. (2021) The datasets have been taken from the Extreme Classification Repository (Bhatia et al., 2016), and preprocessed by doing a tf-idf transformation.

Direct calculation of the unbiased estimate using (77) is prohibitively examples for the small subset of samples in the dataset that have a large number of observed labels (despite the average number of labels per instance being low, some XMC datasets do have outliers with many samples). We can still calculate an unbiased estimate in these cases, at the cost of even further increase in variance: For these examples, we artificially generate subsample observed labels where even more labels have gone missing, and adjust the propensity accordingly. In practice, our implementation divides all propensities by two and drops each ground truth label with a chance of 50% if the ground truth has more than 25 labels. To reduce the increase in variance slightly, we average 100 subsamplings. This technique can be used recursively if the resulting subsample still has too many labels.

In Figure 7, we show histograms for PSRecall estimates at the level of individual samples in the test set.

### C.2    Normalized BCE

In the main text, we claim that the low performance of unbiased training with normalized BCE in the high-regularization regime is due to bad local minima. Here we present supporting evidence.

As shown in Figure 8, the low performance is caused by high loss on training data (dashed lines) as opposed to generalization. However, for training with vanilla loss b), the loss on noisy training data (i.e. the unbiased estimate computed on noisy training data) is much lower than for training with the unbiased loss c), where we directly try to optimize this quantity. There are two possible explanations for this behaviour: Either the switch to the unbiased loss shifts the balance of loss to regularizer such that the regularizer attains much more weight overall, and thus results in less optimization for the loss, or the optimization procedure gets stuck in a local minimum.

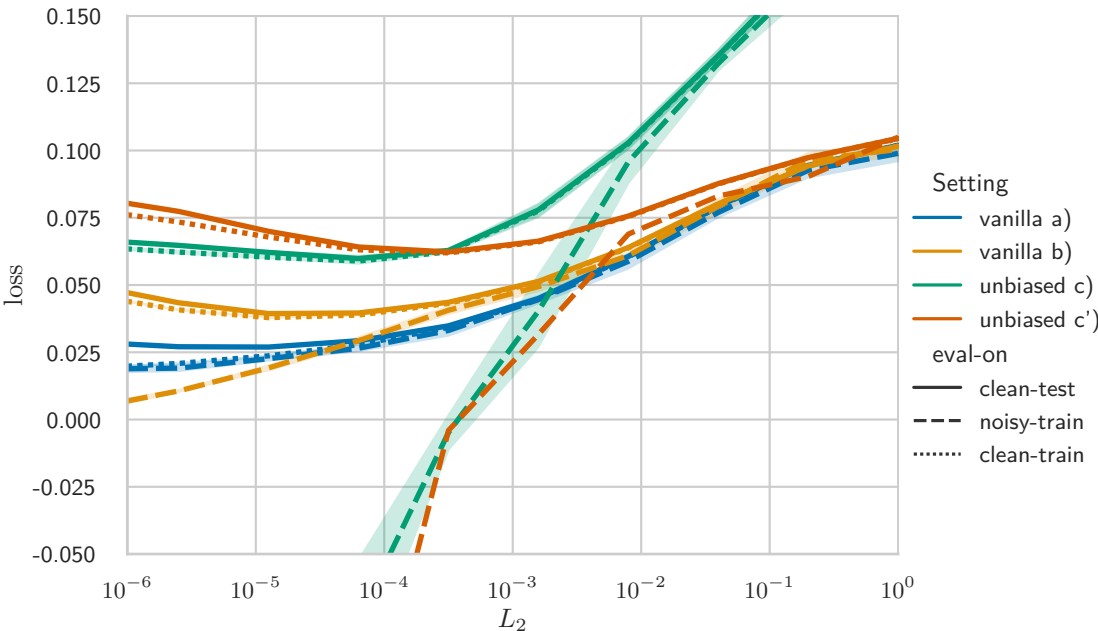

Figure 8: Normalized Binary cross-entropy for different regularization strengths, evaluated on noisy training data, clean training data, and clean test data. Setting c') corresponds to unbiased training with vanilla pre-training.

These two causes can be distinguished by an experiment where the initial weights are chosen by pre-training with vanilla loss for ten epochs. If the reason for the bad performance were the regularization trade-off, then the following 20 epochs of unbiased training would increase the training loss until the trade-off is reached. On the other hand, if the reason were local minima, then starting out close to a known "good" minimum would result in converging to that minimum.

Figure 8 exhibits the behaviour of the second case, thus validating the local-minima hypothesis.

### C.3 Precision and recall comparison figure

Figure 9 shows a graphical representation of the tabular result presented in Table 4.

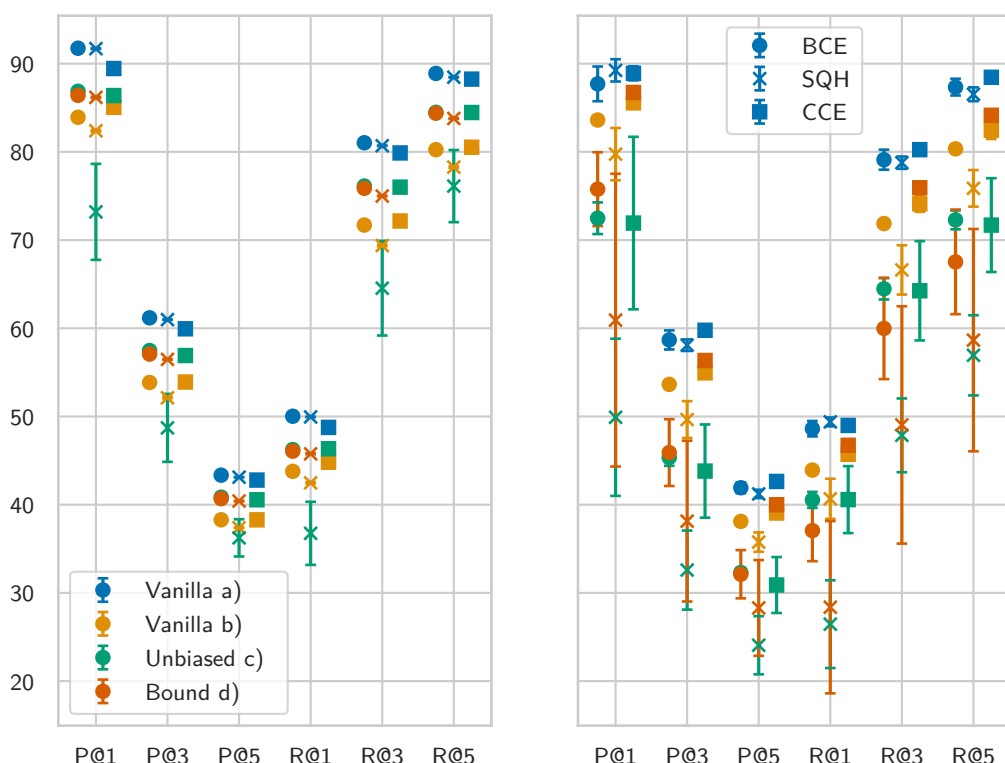

Figure 9: Precision and recall at the optimal (for loss minimization) regularization strength for decomposable (left) and normalized (right) training losses. These values have been calculated on clean test data. The error intervals denote standard deviation and have been determined using 5 runs.

