# OpenReview forum: "Unbiased Loss Functions for Multilabel Classification with Missing Labels"
_TMLR — Accepted by TMLR_

### Review · Reviewer_Busv · 2025-04-23

**Summary Of Contributions:**

This paper investigates the setting of missing labels in binary and multi-label classification with a known missing rate. It points out that few methods have considered the propensity - scored precision, an unbiased estimate for precision-at-k during the training phase, and most methods are limited to loss functions decomposable over labels. However, if the surrogate task is consistent for optimizing recall, the resulting loss function is not decomposable over labels. Hence, this paper develops unbiased estimators for generic, potentially non-decomposable loss functions. Theoretically, it derives new results on uniqueness, variance, and generalization of the unbiased estimators in the binary case, and provides unbiased estimates for general multilabel functions.

**Audience:**

Yes

**Broader Impact Concerns:**

None.

**Claims And Evidence:**

Yes

**Requested Changes:**

1. It is recommended to polish some of the figures.
2. Fix some typos.

**Strengths And Weaknesses:**

- **Strengths**
    1. It introduces a masking model. The formulation of this model can be seen as a generalization of a similar statement given in (Teisseyre et al., 2020) and is convenient for handling expectations in proofs and derivations.
    2. It derives new results on uniqueness (Theorem 10) and variance of the unbiased estimators, and provides a generalization bound (Theorem 9) which is a corrected version of (Natarajan et al., 2017, Lemma 8).
    3. It offers rigorous theoretical derivations for unbiased estimators in binary and multilabel cases, including proofs of existence, uniqueness, etc., which is meaningful.
    4. Experiments are conducted on both synthetic and real data to investigate the proposed unbiased estimator.
    5. The paper is well-structured, with clear logic from problem statement to experimental verification. The use of symbols and the presentation of figures and tables are also intuitive for readers.

- **Weaknesses**
    - The fonts in some images seem inconsistent. For example, the fonts on the horizontal and vertical axes of Figure 3.
    - Some parentheses are used irregularly. For example, on page 22, there are expressions like “(b) for BCE loss" and "settings a)-d)".
    - The question mark "?" in Table 4 is likely to cause misunderstandings. It is recommended to directly mark "lack" instead.

---

> ### Author Response · Authors · 2025-06-06
>
> Thank your for your review. Indeed, it appears that the matplotlib-exported pgf files where selecting a sans-sarif `\\sffamily` front, thus differing in appearance from the rest of the text. We have now manually edited these generated files to remove the font family selection, making the fonts consistent.
>
> We have also changed the notation for alphabetic lists to consistently use a single parenthesis.

---

### Review · Reviewer_8QZi · 2025-04-23

**Summary Of Contributions:**

This paper addresses multilabel classification under missing-label noise, a common issue in extreme multilabel classification (XMC) where positive labels may be missing, but negatives remain unaffected. The authors propose a general framework for constructing unbiased estimators of the true loss, extending beyond decomposable losses to non-decomposable and normalized losses, which are essential for recall-oriented metrics. They establish the theoretical uniqueness of these estimators and highlight the associated bias-variance trade-off, showing that although unbiased estimators correct for bias, they may induce high variance and overfitting. To mitigate this, the paper introduces convex upper bounds as practical surrogates. Experimental results confirm the theoretical insights, particularly regarding the limitations of unbiased estimation when label propensities are low.

**Audience:**

Yes

**Claims And Evidence:**

Yes

**Requested Changes:**

Please carefully checking the weaknesses.

**Strengths And Weaknesses:**

Strengths:

1. The paper offers a comprehensive and rigorous framework for deriving unbiased loss functions under missing-label noise, applicable to both decomposable and non-decomposable multilabel losses.

2. Unlike prior work limited to decomposable losses, this approach covers a wide range of commonly used loss functions, including those aligned with precision and recall metrics.

3. The paper establishes the uniqueness of the proposed unbiased estimators, ensuring theoretical consistency and robustness across different data distributions. It thoroughly explores the trade-off between bias and variance in training under label noise, and connects these insights to generalization behavior.

4. To counteract the instability of high-variance unbiased losses, the paper proposes convex upper-bound surrogates, which are both practically feasible and theoretically justified.

Weaknesses:

1. While theoretically unbiased, the proposed estimators can suffer from extremely high variance, especially under low label propensity, leading to instability and poor generalization in practice.

2. For datasets with very sparse labels or low propensity (common in XMC), the unbiased estimators may become infeasible to compute or unreliable, reducing their applicability in real-world settings.

3. The unbiased estimators for non-decomposable losses require summing over subsets of observed labels, which can be computationally expensive in large label spaces, even with sparsity-based optimizations. It may be beneficial to include a complexity analysis of the proposed method, as this could provide further insight into its practical applicability and scalability.

4. Most empirical studies are on moderately sized datasets or synthetically reduced subsets (e.g., top-100 labels of AmazonCat), which may not fully reflect performance in real extreme-scale XMC settings.

5. The framework assumes access to accurate label propensity estimates, which may be difficult to obtain in many practical scenarios without strong assumptions or additional labeled data.

---

> ### Author Response · Authors · 2025-06-06
>
> Thank your for your review.
>
> Regarding weaknesses 1 and 2, this is an inherent problem of the unbiased estimator, nothing that we can change (esp. with our uniqueness result) :(
> We believe the paper adequately conveys this to the reader.
>
> Reg. 3: you are right, the typical label sparsity assumption in XMC indicates that the _average_ number of positive labels remains low. Due to the exponential cost, however, the unbiased estimator is very sensitive to outliers. We have added an analysis/table that shows the average and maximum number of labels per instance, as well as the required number of terms that would need to be summed up to form the unbiased estimate, on average per sample. This shows that for datasets with seemingly benign average label counts, the cost of the unbiased estimator can still be astronomical. However, if one is willing to filter out (that is, e.g., use a biased estimator) samples with more than 12 labels, on most datasets that would be only about 5%, and then the required cost becomes tractable.
> On some datasets, such as Amazon-3M with its average of 36 labels per point, though, we simply have to concede that the unbiased estimates are untractable.
>
> Reg 4+5:
> We do agree that it would be nice to evaluate the proposed methods real XMC datasets. However, given the lack of sufficiently accurate propensity estimates -- the values commonly used in the XMC literature are crude approximations at best, and evaluating even the benign unbiased decomposable OVA metrics leads to non-sensical results of estimating P@k > 100%, which is only masked in typical XMC literature by normalizing the metric (making it no longer unbiased); you can see this problem illustrated in https://dl.acm.org/doi/pdf/10.1145/3534678.3539466, Table 3.
> We will emphasize this more in the introduction to section 8.
>
> To alleviate this gap, we are currently in the process of running additional experiments with a dataset derived from Yahoo songs R3, which contains a test set that was collected by uniformly random subsampling, which means that we can form sensible propensity estimates for this dataset. We hope to have these additional results ready by early next week.

---

### Review · Reviewer_xYnU · 2025-05-23

**Summary Of Contributions:**

This paper addresses the problem of learning with incomplete annotations in multilabel classification, particularly relevant in extreme multilabel settings where only a sparse subset of relevant labels is available for each instance. The authors:

Present a general mathematical framework to construct unbiased estimators of arbitrary loss functions (including non-decomposable ones) under a known missing-at-random label noise model.

Introduce theoretical guarantees such as uniqueness (Theorem 10, 20) and generalization bounds (Theorem 9), characterizing the inherent bias-variance trade-off in these estimators.

Develop a computationally tractable formulation for these unbiased losses under the masking model (Proposition 4), applicable even in high-dimensional output spaces with sparsely observed labels.

Conduct empirical evaluation demonstrating the limitations of unbiased losses in low-propensity regimes (e.g., high-variance estimators), and showing that in practice, upper-bounded convex surrogates often outperform the unbiased versions.

The paper unifies prior work on unbiased risk estimation and extends it to normalized reductions (for recall@k-consistent training), a previously underexplored setting in extreme multilabel classification.

**Audience:**

Yes

**Claims And Evidence:**

Yes

**Requested Changes:**

Critical
In Section 5.1 (Theorem 19), explicitly quantify computational costs of the proposed unbiased estimators, and clarify in what real-world settings (label sparsity, low class cardinality) they are practically feasible.

Include a small-scale real-world dataset where exact unbiased estimates are computationally tractable and compare against baselines.

Include an ablation comparing upper-bound losses with original unbiased losses on real data to support the bias-variance discussion.

Recommended (Not Critical)
Include more intuitive discussion of Theorem 19: A small worked-out example would help readers understand how the unbiasing operator is computed in practice.

Explore whether data-dependent bounds (Section 6.2, Theorem 25) can be improved, and clarify what conditions would guarantee tightness.

Clarify assumptions in generalization bound (Theorem 9): Explain how realistic the assumptions are in the context of XMC tasks—especially the Lipschitz continuity and boundedness of the loss functions.

Figure 2 could include clearer labeling and error bars for easier visual interpretation. Also clarify in caption what “vanilla” means.

Several important results (e.g., Theorem 20, Lemma 21) refer readers to the appendix. Consider adding brief outlines or proof sketches in the main text to improve readability.

**Strengths And Weaknesses:**

Strengths
The paper introduces well-formalized definitions and theorems, such as compatibility conditions, masking models, and the general unbiasing operator for multilabel losses (Theorem 19).

A significant contribution, enabling unbiased training for recall-consistent objectives (as opposed to only precision-consistent ones).

The empirical analysis highlights when unbiased estimators are effective versus when high variance renders them unusable.

The theoretical and empirical study of variance growth in low-propensity settings is well-motivated and adds useful understanding to practical applications.

The paper bridges the gap between PU-learning, propensity scoring, and unbiased multilabel loss estimation in a coherent way.

Weaknesses

Despite efforts to limit the cost, the unbiased estimator for non-decomposable losses involves exponential complexity in label cardinality (though mitigated in sparse settings). This may limit scalability.

Experiments are limited to synthetic or modified datasets. Evaluation on full-scale XMC benchmarks (e.g., Amazon-670K, WikiLSHTC) is absent due to the infeasibility of calculating the unbiased estimates—ironically undermining the full practical relevance.

While convex upper bounds are useful in practice, the derivation often appears heuristic (e.g., replacing
$$Y^*$$ with $$Y/p$$ in normalizations), and lacks tightness guarantees.

Some readers may find the use of measure-theoretic language and intricate definitions (like compatibility or masking model) unnecessarily heavy for the intended ML audience.

---

> ### Author Response · Authors · 2025-06-06
>
> Thank you for your thorough review.
> We have added a table next to Thm 19 that shows that average label sparsity is a necessary, but not sufficient condition for computational tractability. In particular, outlier instances with a very large number of positives are problematic.
>
> We are working on a small scale (ca 500 labels) example based on the Yahoo-Music R3 data, where the test set has been uniformly sampled, allowing us to determine propensities directly. Unfortunately, this is not quite ready yet - we expect to have results early next week.
>
> In the appendix, we have an example that shows how Thm 19 is used to derive a formula for unbiased recall calculation.
>
> Both Theorem 20 and Lemma 21 already have brief proof sketches in the paper.

---

> > ### Author Response · Authors · 2025-06-11
> >
> > We attempted to use the data mentioned above for additional experiments. While the results generally agree with the synthetic data experiment already in the paper, they do so in a rather uninteresting way:
> > As expected, at higher regularization strength the unbiased estimator/bound outperforms the original loss function. However, all three losses appeared to perform best when the regularization is extremely large. This means that apparently, the best predictor that the classifier could find is the constant one, and using the unbiased loss just helps in picking the right labels to always predict.
> > Maybe this isn't too surprising, given the extreme noise level in the data; on the test set, only 1% of all labels will be present, meaning that even for a song that 50% of all 5000 listeners would like, only 25 positive instances could be found in the test set.
> > Additionally, the different sampling strategies for training and test set will also lead to a distribution shift in the features, further making learning more difficult.
> >
> > If you have any other suggestions for multilabel datasets to use where the it is possible to get accurate propensity estimates, we'd be happy to hear them.
> > We'd like to point out that the [multilabel reductions paper](https://papers.nips.cc/paper_files/paper/2019/hash/da647c549dde572c2c5edc4f5bef039c-Abstract.html) only validates its approach on very simple synthetic data (mixture of Gaussians).

---

### Decision · Action_Editor_7XVc · 2025-07-11

**Recommendation:** Accept with minor revision

**Additional Comments:**

There were discussions about additional experiments with Yahoo-Music R3 data during the discussion phase. Please add this experiment to the camera-ready version of the paper.

**Audience:**

Yes

**Audience Explanation:**

Some in TMLR's audience are working on supervised learning, multilabel classification, and extreme multi-label classification, and I expect they will be interested in the findings of this paper.

**Claims And Evidence:**

Yes

**Claims Explanation:**

This paper addresses extreme multilabel classification in settings where the positive labels may be missing due to the large number of possible labels. It makes a series of contributions by showing that the modeling of missing labels by using a mask variable provides a way to derive unbiased estimators under some assumptions, studies uniqueness and variance of the unbiased estimators, and provides generalization bounds. Due to the unboundedness of the estimator, the paper proposes to use convex upper-bounds and validate them through theoretical analysis and experiments. All reviewers appreciate the novelty and contributions of the work. The reviewers pointed out a few concerns, but the rebuttal provided additional results with more explanation, intuition, experiments, and improved the presentation and clarity. Finally, all reviewers recommended acceptance.